# Molecular recognition of an odorant by the murine trace amine-associated receptor TAAR7f

Anastasiia Gusach [1], Yang Lee [1], Armin Nikpour Khoshgrudi[2,3], Elizaveta Mukhaleva[4], Ning Ma [4], Eline J. Koers[2,3], Qingchao Chen[1], Patricia C. Edwards[1], Fanglu Huang [5], Jonathan Kim[6], Filippo Mancia[6], Dmitry B. Veprintsev [2,3], Nagarajan Vaidehi [4], Simone N. Weyand [5,7,8,9] & Christopher G. Tate [1]

There are two main families of G protein-coupled receptors that detect odours in humans, the odorant receptors (ORs) and the trace amine-associated receptors (TAARs). Their amino acid sequences are distinct, with the TAARs being most similar to the aminergic receptors such as those activated by adrenaline, serotonin, dopamine and histamine. To elucidate the structural determinants of ligand recognition by TAARs, we have determined the cryo-EM structure of a murine receptor, mTAAR7f, coupled to the heterotrimeric G protein $G_s$ and bound to the odorant N,N-dimethylcyclohexylamine (DMCHA) to an overall resolution of 2.9 Å. DMCHA is bound in a hydrophobic orthosteric binding site primarily through van der Waals interactions and a strong charge-charge interaction between the tertiary amine of the ligand and an aspartic acid residue. This site is distinct and non-overlapping with the binding site for the odorant propionate in the odorant receptor OR51E2. The structure, in combination with mutagenesis data and molecular dynamics simulations suggests that the activation of the receptor follows a similar pathway to that of the β-adrenoceptors, with the significant difference that DMCHA interacts directly with one of the main activation microswitch residues, Trp[6.48].

Perception and interpretation of odours are essential for the life of vertebrates. Odorant molecules are detected in the nasal cavity by G protein-coupled receptors (GPCRs) in olfactory sensory neurons, which then transmit a signal to the olfactory bulb in the brain[1]. Each olfactory sensory neuron specifically expresses a single chemosensory GPCR that is activated by one or several volatile odorant molecules[2–5]. Detection of thousands of different odours is then possible through the action of hundreds of receptor types in their corresponding neurons. Approximately half of the ~800 human GPCRs are chemosensory receptors[6]. These belong to the rhodopsin-like class A family and can

[1]MRC Laboratory of Molecular Biology, Francis Crick Avenue, Cambridge CB2 0QH, UK. [2]Centre of Membrane Proteins and Receptors (COMPARE), University of Birmingham and University of Nottingham, Midlands NG7 2RD, UK. [3]Division of Physiology, Pharmacology and Neuroscience, School of Life Sciences, University of Nottingham, Nottingham NG7 2UH, UK. [4]Department of Computational and Quantitative Medicine, Beckman Research Institute of the City of Hope, 1218 S 5th Ave, Monrovia, CA 91016, USA. [5]Department of Biochemistry, University of Cambridge, Tennis Court Road, Cambridge, UK. [6]Department of Physiology and Cellular Biophysics, Columbia University Irving Medical Center, New York, NY 10032, USA. [7]Department of Medicine, University of Cambridge, Victor Phillip Dahdaleh Building, Heart & Lung Research Institute, Papworth Road, Cambridge Biomedical Campus, Cambridge CB2 0BB, UK. [8]Cambridge Institute for Medical Research, Keith Peters Building, Biomedical Campus, Hills Rd, Cambridge CB2 0XY, UK. [9]EMBL's European Bioinformatics Institute (EMBL-EBI), Wellcome Genome Campus, Hinxton, Cambridgeshire CB10 1SD, UK. ✉e-mail: sw644@cam.ac.uk; cgt@mrc-lmb.cam.ac.uk

be divided into two groups, the odorant receptors (ORs) and the trace amine-associated receptors (TAARs). Although rhodopsin-like GPCRs are the most abundant receptors in humans and the most well-studied from the structural and functional perspectives[7,8], only recently has the first structure of an OR been determined[9].

The TAARs are a small family of specialised receptors with only seven representatives encoded by the human genome, compared to about 400 OR genes. TAARs are most similar to aminergic receptors, such as the $\beta_2$-adrenoceptor ($\beta_2$AR), serotonin 5-HT$_4$ receptor, dopamine D$_1$ receptor and histamine H$_2$ receptor, with amino acid identity between these four receptors and hTAAR9 varying between 27–32% (Supplementary Fig. 1a). In comparison, hTAAR9 is only 11–16% identical to a range of human ORs. TAARs bind volatile amines[3,4] that are typically small molecules formed by the decarboxylation of amino acids[10]. These molecules serve as sensory cues for a range of stimuli[10–13], such as the presence of predators or prey, the proximity of a mating partner and the spoilage of food, and elicit either attraction or aversion responses, depending on the odour and its intensity. The TAAR receptor studied here, mTAAR7f, has well-characterised agonists[14,15] and its closest human homologue based on amino acid sequence is hTAAR9 (sequences are 71% identical; Supplementary Fig. 1b), implying significant conservation of their structures. Mice exhibit either attractive/neutral or aversive behaviour when exposed to TAAR7f ligands, such as amines found in urine[16], with the physiological response dependent upon the ligand and its concentration[14].

## Results

### Structure determination of TAAR7f

In order to define the molecular recognition determinants of odorants by TAAR receptors, we determined a structure of mTAAR7f by electron cryo-microscopy (cryo-EM) in an active state coupled to the heterotrimeric G protein, G$_s$. mTAAR7f was chosen from a screen of multiple different olfactory receptors as being a highly expressed receptor in

insect cells using the baculovirus expression system and also being relatively stable after detergent solubilisation as assessed by fluorescence-detection size exclusion chromatography (FSEC). In addition, TAAR7f is known to bind the agonist N,N-dimethylcyclohexylamine (DMCHA) with an EC$_{50}$ of 0.5 μM[15].

Wild-type mTAAR7f was tagged at the N-terminus (haemagglutinin signal sequence, FLAG tag, His$_{10}$ purification tag and tobacco etch virus cleavage site) and C-terminus (human rhinovirus 3C cleavage site and eGFP). The construct was expressed in insect cells using the baculovirus expression system and purified in the presence of the agonist DMCHA (see 'Methods'; Supplementary Fig. 2a–c). In vivo, mTAAR7f couples to the heterotrimeric G protein G$_{olf}$[17]. However, G$_{olf}$ and G$_s$ have very similar amino acid sequences (77% identical) and the molecular determinants of G$_s$ coupling identified in GPCR-G$_s$ structures[18] were predicted to be identical in G$_{olf}$, assuming that the respective G proteins couple in a similar fashion. We therefore used G$_s$ for making a mTAAR7f complex, because of the availability of nanobody Nb35 that stabilises the interface between the α-subunit and β-subunit of the heterotrimeric G protein; this may have proved important in improving the stability of the complex[19].

Purified DMCHA-bound mTAAR7f was mixed with mini-G$_s$ heterotrimer and Nb35 to form a complex that was isolated from the unbound G protein by size exclusion chromatography (Supplementary Fig. 2a–c) and vitrified for single-particle cryo-EM analysis (Supplementary Fig. 2d, e). The reconstruction of the mTAAR7f-G$_s$ complex (Fig. 1a, b) had a nominal resolution of 2.9 Å (Supplementary Fig. 2f, g and Supplementary Table 1). The receptor portion of the complex was flexible and therefore focussed refinement was used to improve its resolution from 3.5 Å to 3.2 Å (Supplementary Figs. 2f, g and 3). The density of the ligand was clearly distinguishable (Fig. 1b) and the planar configuration of the ligand was observed by the density's flattened oval shape. The ligand was placed within this density with the positively charged tertiary amine group adjacent to the carboxylate of Asp127[3.32] (superscript

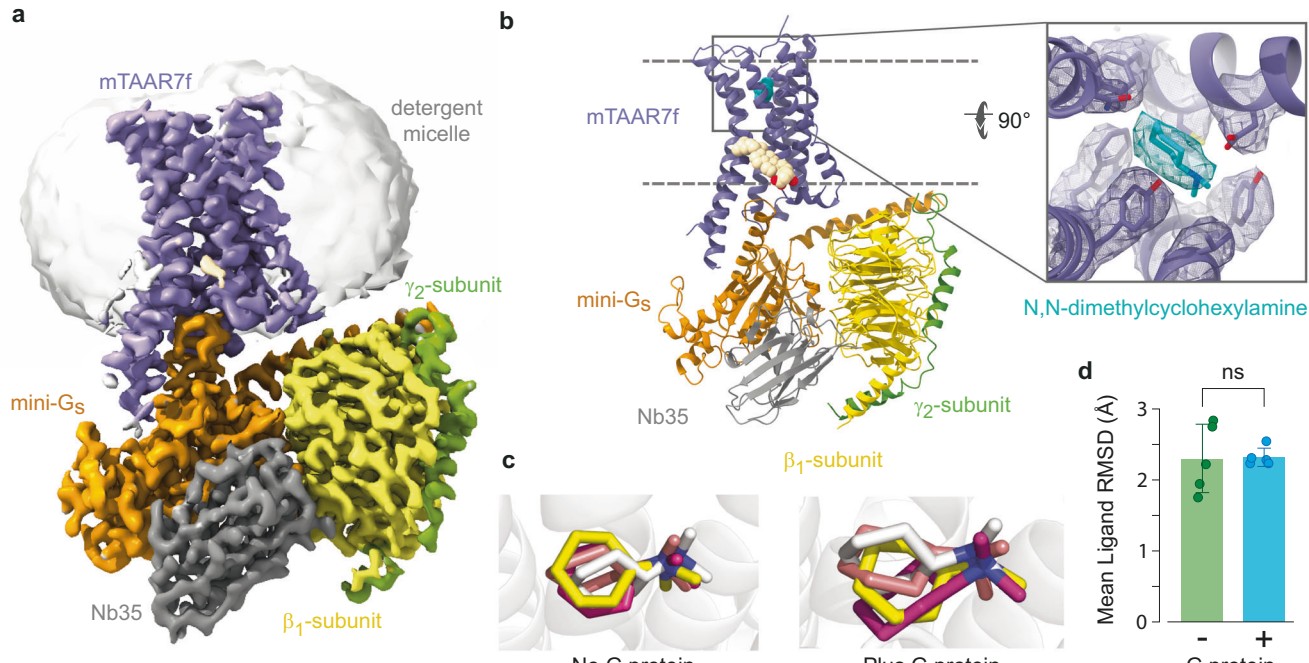

**Fig. 1 | Overall structure of the mTAAR7f-G$_s$ complex. a** Cryo-EM density of the entire complex. **b** Cartoon of the mTAAR7f-G$_s$ complex (ribbon representation) with bound DMCHA (pale blue) and CHS (pale brown) shown as spheres. The inset shows density for DMCHA (pale blue) and surrounding residues (purple) in mesh. The view is from the extracellular surface and is 90° orthogonal to the receptor cartoon viewed in the membrane plane. **c** The three most populated ligand binding

poses derived from MD simulations conducted either in the absence or presence of the G protein (ligand orientation from the cryo-EM structure is shown in grey) are shown in stick representation. **d** Differences in ligand RMSD from five distinct MD simulations either with or without G protein are depicted. The error bars represent the SD and a two-sided $t$-test showed no statistical difference (ns; $p = 0.94$) between the mean ligand RMSDs.

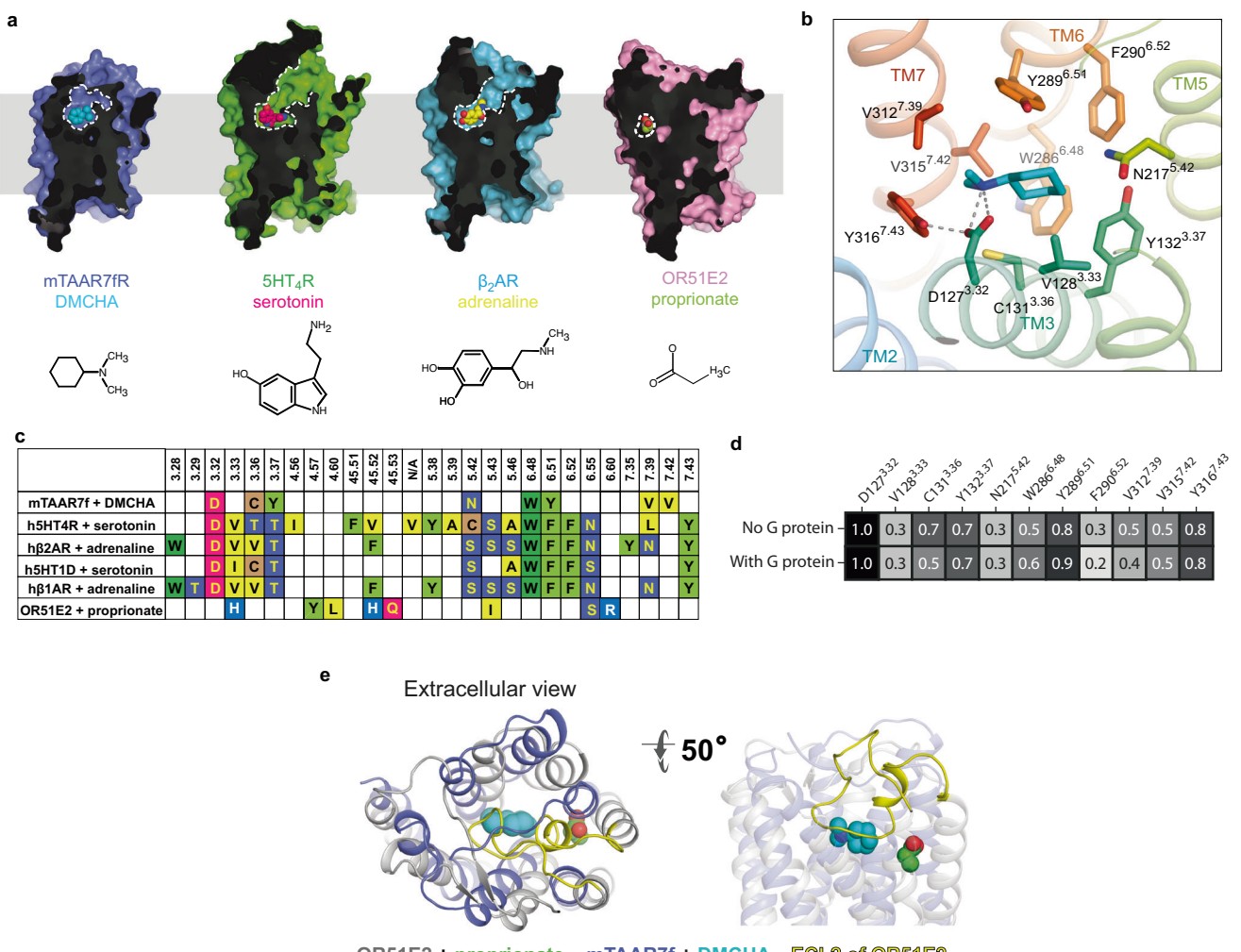

**Fig. 2 | The mTAAR7f orthosteric binding site and comparison to other receptors. a** Sliced surface representation of the OBS of DMCHA-bound mTAAR7f, serotonin-bound 5HT₄R, adrenaline-bound β₂AR and propionate-bound OR51E2; ligand atoms are depicted as spheres and the ligand structures are shown at the bottom of the panel. **b** Binding pose of DMCHA and details of ligand-receptor interactions. Amino acid residues ≤3.9 Å from the ligand are shown with polar interactions depicted as dashed lines. **c** Amino acid residues in the OBS within 3.9 Å of the ligand of mTAAR7f, aminergic receptors and an odorant receptor, OR51E2: (PDB IDs; h5HT₄R, 7XT8; β₁AR, 7JJO; β₂AR, 4LDO; h5HT₁DR, 7E32; OR51E2, 8F76). Superscripts refer to the Ballesteros–Weinstein naming convention[28]. **d** Frequency of ligand contacts as determined during MD simulations. **e** The relative positions of the OBS in OR51E2 and mTAAR7f are shown after superposition of the receptors. Ligands (propionate and DMCHA) are shown as spheres.

refs to the Ballesteros–Weinstein numbering system). This position was corroborated by molecular dynamics (MD) simulations (Fig. 1c, d).

## Overall architecture of mTAAR7f and the orthosteric binding site

The overall structure of mTAAR7f resembles the canonical structure of Class A GPCRs coupled to a G protein. Two of the most closely related receptors to mTAAR7f by amino acid sequence are β₂AR and the serotonin 5-HT₄ receptor (5-HT₄R). Comparison of the active state structure of mTAAR7f with G protein-coupled structures of β₂AR and 5-HT₄R showed considerable similarity (all Cα atoms RMSDs of 1.9 Å and 2.7 Å, respectively). In contrast, there is little similarity in structure between mTAAR7f and the only other olfactory receptor structure OR51E2[9] (RMSD of 5.0 Å, all Cα atoms), which is also in an active state coupled to mini-Gₛ.

The position of the orthosteric binding site (OBS) resembles closely that of the aminergic receptors and not that of OR51E2 (Fig. 2a). In mTAAR7f, the agonist DMCHA is found in a cavity formed by transmembrane helices TM3, TM5, TM6 and TM7, and is separated from the outside of the cell by extracellular loop (ECL) 2 (Fig. 2a, e), which is held

in position across the OBS by the Class A canonical disulfide bond between Cys205ECL2 and Cys120[3.25]. The OBS of mTAAR7f overlaps the positions of the OBS in 5-HT₄R and β₂AR and the position of the agonists also overlap (Fig. 2a, c), but the pocket itself is smaller and lacks the extracellular access seen in the aminergic receptors. In contrast, the even smaller binding pocket of propionate in the OR51E2 structure and the position of the agonists do not overlap at all with mTAAR7f, despite sharing the same occluded architecture.

All the receptor-ligand contacts (≤3.9 Å; Fig. 2b, c) in mTAAR7f are mediated by eight amino acid residue side chains, three of which are aromatic (Tyr132[3.37], Trp286[6.48], Tyr289[6.51]), three hydrophobic (Cys131[3.36], Val312[7.39], Val315[7.42]) and two polar (Asp127[3.32], Asn217[5.42]). All of the interactions are mediated by van der Waals interactions with the exception of a strong polar interaction between the charges on Asp127[3.32] and the tertiary amine in DMCHA. In MD simulations (Fig. 2d), this interaction was preserved 100% of the time (5 simulations, 1 μs each, 50,000 snap shots per simulation). Other receptor-ligand interactions identified in the cryo-EM structure mediated by van der Waals contacts are present 30–90% of the time (Fig. 2d). In addition, the MD simulations identified three other residues that make

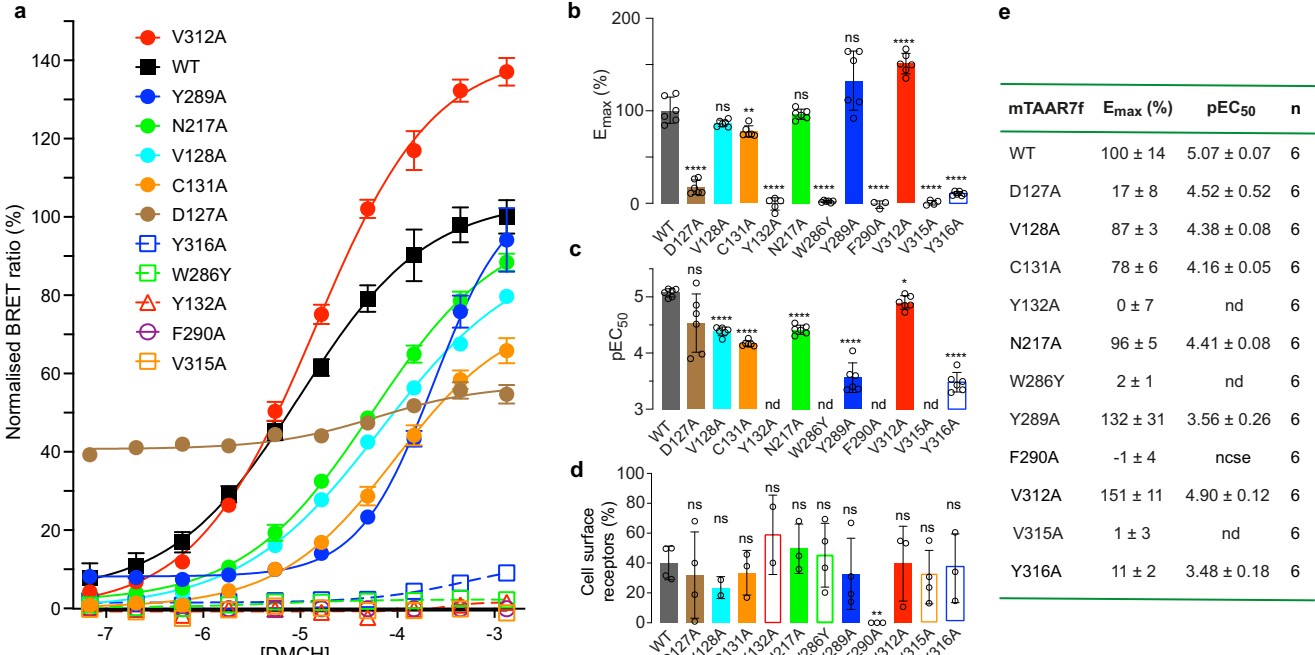

**Fig. 3 | G protein recruitment to mTAAR7f. a** BRET ratios measured for increasing concentrations of the agonist DMCHA for the wild-type mTAAR7f (WT) and 11 mutants, and normalised with respect to cell surface expression levels. Three to six experiments were performed independently (see Supplementary Table 2 for details) with single measurements per experiment and error bars represent the SD. **b** Comparison of values for $E_{max}$. **c** Comparison of values for $EC_{50}$. **d** Cell surface expression data. In (**b**–**d**), error bars represent the SD with the statistical significance compared to the wild type receptor determined using a two-sided Welch's $t$-test (****$p < 0.0001$; **$p$ 0.001 to 0.01; *$p$ 0.01 to 0.05; ns not significant, $p \geq 0.05$). The number of independent experiments and numerical values for the mean, errors and $p$ values are given in Supplementary Table 2. **e** Numerical values determined from data plotted in (**b**) and (**c**).

contact to the ligand 20–80% of the time (Val128[3.33], Phe290[6.52], Tyr316[7.43]; Fig. 2d) that are as close as 4.1 Å, 4.0 Å and 5.2 Å (respectively) to DMCHA in the cryo-EM structure. Mutagenesis of residues predicted to make contact to the ligand (either from the cryo-EM structure or from MD simulations) in most cases significantly decreased the ability of G protein to be coupled (decreased pEC50; Fig. 3a–e). No coupling was observed for the mutants Y132A[3.37], W286Y[6.48] and V315A[7.42] despite the presence of the receptor at the cell surface. Significant decreases in pEC50 were also observed for V128A[3.33], C131A[3.36], N217A[5.42], Y289A[6.51] and Y316A[7.43]. The role of Phe290[6.52] in ligand binding could not be assessed as the mutant F290A[6.52] was not expressed at the cell surface; Phe290[6.52] appears to play a structural role by making van der Waals contacts with Phe218[5.44] and Phe221[5.47] that form part of a column containing five phenylalanine residues holding the extracellular ends of TM5 and TM6 together. Mutation of Asp127[3.32] also had a dramatic effect on G protein coupling, with the mutant D127A[3.32] having increased basal activity, decreased $E_{max}$ and a similar pEC50 to the wild type receptor. The increased basal activity may be due to destabilisation of the inactive state, as the hydrogen bond between Asp127[3.32] and Tyr316[7.43] would be broken, although there could be other unknown effects of this mutation on the dynamics of the receptor that also contribute. Of the residues in the OBS of mTAAR7f, Asp127[3.32], Trp286[6.48] and Tyr316[7.43] are all absolutely conserved in all murine and human TAARs (Supplementary Fig. 4b).

Comparison between residues involved in receptor-ligand contacts in mTAAR7f and the aminergic receptors β2AR, 5-HT4R and 5-HT1R highlight commonalities and differences. Two conserved residues (Asp127[3.32], Trp286[6.48]) make contacts to the respective agonists in all four structures (Fig. 2c), with an additional four residues always making contacts (positions 3.36, 3.37, 5.42, 6.51) and one residue often making contacts (position 7.39). The interaction between DMCHA and Asp127[3.32] is particularly striking as this residue makes interactions with a nitrogen atom in ligands binding to GPCRs throughout the aminergic

family (Supplementary Fig. 4a). ECL2 often makes contacts to ligands in aminergic receptors (Supplementary Fig. 4a), but does not make contacts to DMCHA in mTAAR7f. In the structure of OR51E2, the agonist propionate makes contacts to eight amino residue side chains that form a binding site with strong polar attributes due to the presence of five polar side chains (His104[3×33], His180[45×52], Gln181[45×53], Ser258[6×55], Arg262[6×59]) and only three hydrophobic side chains (Phe155[4×57], Leu158[4×60], Ile202[5×43])[9]. This is distinct from the predominantly hydrophobic OBS in mTAAR7f. None of the ligand-binding residues in OR51E2 correlate with ligand-binding residues in mTAAR7f, although four of the residues (positions 3×33, 45×52, 5×43 and 6×55) correspond to ligand-binding residues in aminergic receptors, including a residue from ECL2 (Fig. 2c). MD simulations of mTAAR7f indicate that the ECLs are dynamic and allow rapid binding of DMCHA (within 100–200 ns, four out of five trajectories (see 'Methods'; Supplementary Fig. 5a, b), but none of the ECL residues are involved in interacting with the ligand in any of its lowest energy states.

Previous structure-activity relationship (SAR) data for mTAAR7f suggests that ligand binding is highly dependent on the ligand shape and the length of the hydrophobic chain, with aliphatic chains containing less than six carbon atoms being unable to activate the receptor[15]. The size and shape of the OBS seen in the cryo-EM structure clearly imposes restrictions on which ligands can bind. The mTAAR7f mutant Y132C[3.37] was predicted to expand the size of the OBS and to allow binding of bulkier ligands that activate mTAAR7e which contains a Cys residue at this position; the mutation did indeed reverse the ligand selectivity of the two receptors as predicted[15] and is consistent with the cryo-EM structure, as a smaller residue at this position would allow ligands to pack between TM3 and TM5.

## G protein coupling interface
The position of the heterotrimeric mini-Gs protein in relation to mTAAR7f is similar to that in other class A GPCRs[18]. However,

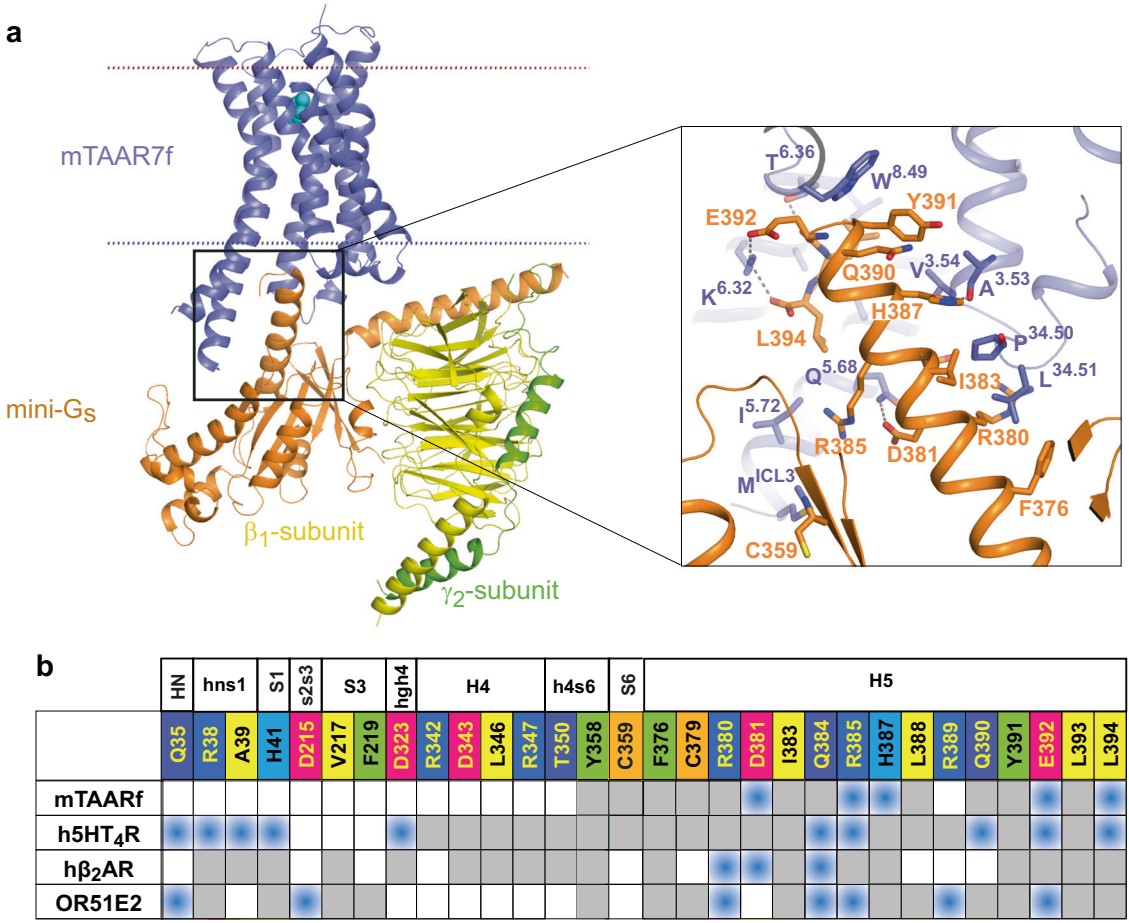

**Fig. 4 | Interactions between mini-G$_s$ and mTAAR7f. a** Cartoon of the mTAAR7f-G$_s$ complex with an inset highlighting interactions between the α5 helix of mini-G$_s$ and mTAAR7f (distance cut-off ≤3.9 Å). **b** Comparison of amino acid contacts (distance cut-off ≤3.9 Å) made by the α-subunit of G$_s$ and mTAAR7f, h5HT$_4$R (PDB 7XT8), hβ$_2$AR (PDB 3SN6) and OR51E2 (PDB 8F76); blue, polar contacts; grey, van der Waals contacts.

| | HN | hns1 | | | S1 | s2s3 | S3 | | hgh4 | H4 | | | | h4s6 | S6 | H5 | | | | | | | | | | | | | | |
|---|---|---|---|---|---|---|---|---|---|---|---|---|---|---|---|---|---|---|---|---|---|---|---|---|---|---|---|---|---|---|
| | Q35 | R38 | A39 | H41 | D215 | V217 | F219 | D323 | R342 | D343 | L346 | R347 | T350 | Y358 | C359 | F376 | C379 | R380 | D381 | I383 | Q384 | R385 | H387 | L388 | R389 | Q390 | Y391 | E392 | L393 | L394 |
| **mTAARf** | | | | | | | | | | | | | | | | | | | ● | | ● | ● | ● | | | ● | | ● | | |
| **h5HT$_4$R** | ● | ● | ● | ● | | | | ● | | | | | | | | | | ● | | | ● | ● | ● | | | ● | | | | |
| **hβ$_2$AR** | | | | | | | | | | | | | | | | | | ● | ● | | ● | | | | | | | | | |
| **OR51E2** | ● | | | | ● | | | | | | | | | | | | | ● | ● | | ● | ● | | ● | | | | | | |

compared to its nearest homologues β$_2$AR and 5-HT$_4$R, and to OR51E2, mTAAR7f forms fewer contacts between the receptor and the α-subunit of the G protein (Fig. 4a, b). Seventeen residues in the α-subunit make contacts to the receptor, thirteen of which are in the α5 helix and are conserved in other G$_s$-coupled receptor structures. The amino acid identity of all the α-subunit residues in contact with the receptor are 100% conserved between G$_s$ and G$_{olf}$. The area of the mTAAR7f-G$_s$ interface is smaller (1140 Å$^2$) compared to that in the 5-HT$_4$R-G$_s$ (1580 Å$^2$) or β$_2$AR-G$_s$ (1260 Å$^2$) complexes.

## Ligand-induced activation of mTAAR7f

The activation of class A GPCRs by diffusible ligands occurs through a series of structural changes commencing with agonist binding, followed often by a contraction of the OBS and then propagation of structural changes through the receptor to the intracellular G protein binding interface (Fig. 5a, b). The resulting outward shift of the intracellular end of TM6 enables coupling of a G protein, as exemplified by the active state of the β$_2$AR[19]. An alignment between mTAAR7f and β$_2$AR is shown in Supplementary Fig. 6 to facilitate the discussion below. The orientation of mTAAR7f transmembrane helices in the cryo-EM structure aligns well with the G protein-coupled active state of β$_2$AR (Fig. 5a, b). In addition, mTAAR7f contains hallmarks of activation in the conserved regions essential for stabilisation of the active state, including the P$^{5.50}$-I$^{3.40}$-F$^{6.44}$ motif, the C$^{3.36}$-W$^{6.48}$-x-F$^{6.44}$ motif, the D$^{3.49}$-R$^{3.50}$-Y$^{3.51}$ motif, and the N$^{7.49}$P$^{7.50}$xxY$^{7.53}$ motif (Fig. 5a and Supplementary Fig. 4c). The ionic lock between Arg$^{3.50}$ and Asp$^{3.49}$ is a hallmark of an inactive state of Class A receptors, which is broken upon receptor activation through a rotamer change of Arg$^{3.50}$. In the mTAAR7f structure, the positions of Arg145$^{3.50}$ and Asp144$^{3.49}$ are identical to the equivalent residues in the active state of β$_2$AR (Fig. 5a). Similarly, the positions in mTAAR7f of Tyr326$^{7.53}$ in the NPxxY motif and the associated Tyr232$^{5.58}$ align well with the equivalent residues in the active state of β$_2$AR and not the inactive state. However, only portions of the CWF and PIF motifs follow the canonical pattern of rotamer conformations observed in β$_2$AR (Fig. 5a). Phe282$^{6.44}$ in the PIF motif in mTAAR7f does align well with the respective rotamer in β$_2$AR, but Leu135$^{3.40}$ in mTAAR7f cannot adopt the active conformation of Ile$^{3.40}$ in β$_2$AR due to the position of Trp286$^{6.48}$. The position of Trp286$^{6.48}$ in mTAAR7f is rotated by 35˚ around the TM6 helical axis compared to its position in β$_2$AR, resulting in a 4.2 Å difference in its position (measured at the CH2 atom). The shift of Trp286$^{6.48}$ in mTAAR7f also causes Phe$^{6.44}$ to adopt an active state conformation to prevent a clash. The position of the highly conserved Trp$^{6.48}$ in Class A GPCRs has been described as one of the key elements of activation of many GPCRs[20], making this a likely candidate in the activation of mTAAR7f.

Why does Trp286$^{6.48}$ adopt such an extreme conformation compared to β$_2$AR? The ligand DMCHA makes van der Waals contacts to Trp286$^{6.48}$ and this could be one reason why it is shifted greatly compared to its position in β$_2$AR. The rotations of DMCHA observed in the MD simulations would place Trp286$^{6.48}$ in this position and this is evident from the Chi2 dihedral angle fluctuations being far lower when DMCHA is bound compared to when it is not (compare Step 2 with Step 3 in the ligand binding pathway, Supplementary Fig. 5a). The rotamer of Trp286$^{6.48}$ is directly impacted by the DMCHA and is the last

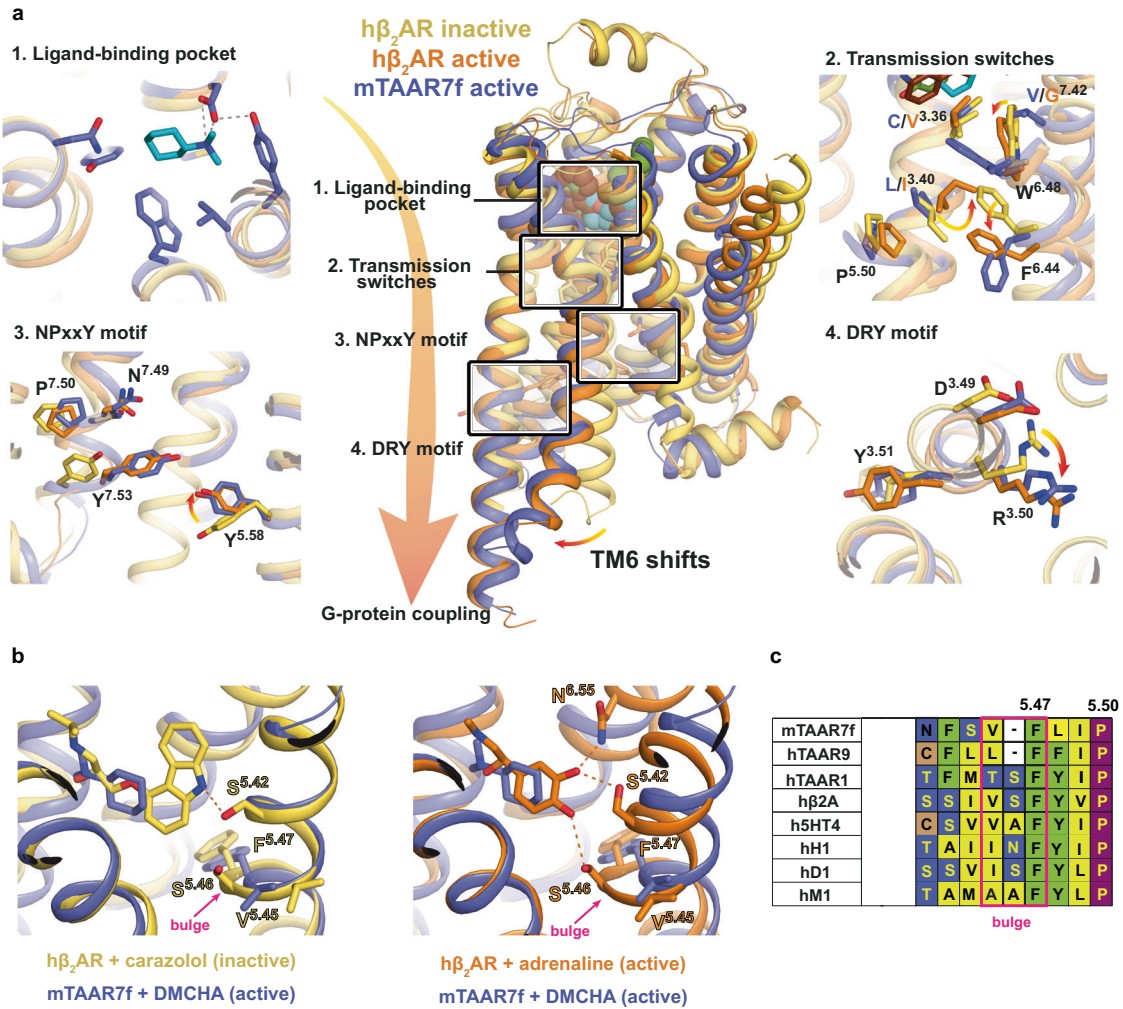

**Fig. 5 | Activation switches in mTAAR7f and β₂AR. a** Conformational changes in functional motifs are depicted in an alignment of the inactive state structure of hβ₂AR (yellow, carazolol-bound, PDB 2RH1), an active state structure of hβ₂AR (orange, BI-167107-bound, PDB 3SN6) and the mTAAR7f structure (purple). **b** Increase in the TM5 bulge in β₂AR upon the transition from an inactive state (left panel, yellow, PDB 2RH1) to the active state (right panel, orange, PDB 4LDO). Both structures are aligned with the active structure of mTAAR7f-Gs-DMCHA (purple). Hydrogen bonds between the receptors and their corresponding ligands are shown as dashed lines. **c** Alignment of amino acid residues in the bulge region of aminergic GPCR representatives with mTAAR7f, hTAAR9 and hTAAR1. One amino acid in the bulge region is absent in mTAAR7f and hTAAR9.

step in the ligand binding process observed by MD (Step 4, Supplementary Fig. 5a). Another residue that may play a role in the position of Trp286[6.48] is Val315[7.42] which is only 4 Å from Trp286[6.48] in mTAAR7f, and would clash if Trp286[6.48] were to adopt the active state conformation observed in β₂AR. The importance of these two residues in the activation of mTAAR7f is apparent from mutagenesis data. The mutants W268Y[6.48] and V315A[7.42] both show significantly decreased $E_{max}$ and $EC_{50}$ for activation compared to the wild type receptor (Fig. 3a–e).

Comparisons between mTAAR7f and β₂AR can also help to formulate a mechanism of how DMCHA binding may potentially activate the receptor. In β₁AR and β₂AR, ligand-induced activation is caused by the para-hydroxyl of the catecholamine moiety of the agonist inducing a rotamer change of Ser[5.46] and the contraction of the OBS by 1–2 Å[21,22]. Coupling of the G protein causes a further contraction of the OBS, predominantly through the movement of the extracellular ends of TM6 and TM7, resulting in decreased on/off rates of the ligand and an increase in agonist affinity due to an increased number and/or strength of ligand-receptor interactions[23,24]. mTAAR7f differs from the βARs in that there is only a weak van der Waals interaction between the agonist and TM5 (Asn217[5.42]), and also that the characteristic bulge formation upon ligand activation of βARs is absent. Amino acid sequence

alignments between aminergic receptors and TAARs show that there is a one amino acid deletion in this region in the TAARs (Fig. 5c), leading to TM5 being unable to form a bulge. Therefore, it is likely that the activation cascade upon ligand binding to mTAAR7f differs subtly to that of the βARs.

Based on the active-state mTAAR7f structure and extensive knowledge of the activation of the βARs, we suggest here a possible mechanism of ligand activation of m7TAARf. Binding of DMCHA occurs predominantly through charge-charge interactions between the tertiary amine of the ligand and Asp127[3.32], and extensive van der Waals interactions with the hydrophobic OBS. This causes a contraction of the OBS through interactions between DMCHA and residues in TM6 and TM7 (Val312[7.39], Val315[7.42], Y289[6.51], Trp286[6.48]) and the stabilisation of the interaction between TM3 and TM7 by a hydrogen bond between Tyr316[7.43] and Asp127[3.32]. The position of the DMCHA and Val315[7.42] causes Trp286[6.48] to rotate and induce activation of downstream motifs (PIF, NPxxY and DRY), ultimately resulting in the outward movement of TM6 and G protein coupling. Of course, in the absence of an inactive state structure of mTAAR7f this is currently a working hypothesis, but it is supported by both mutagenesis data and MD simulations. The mutants W286Y[6.48] and Val315[7.42] both show low

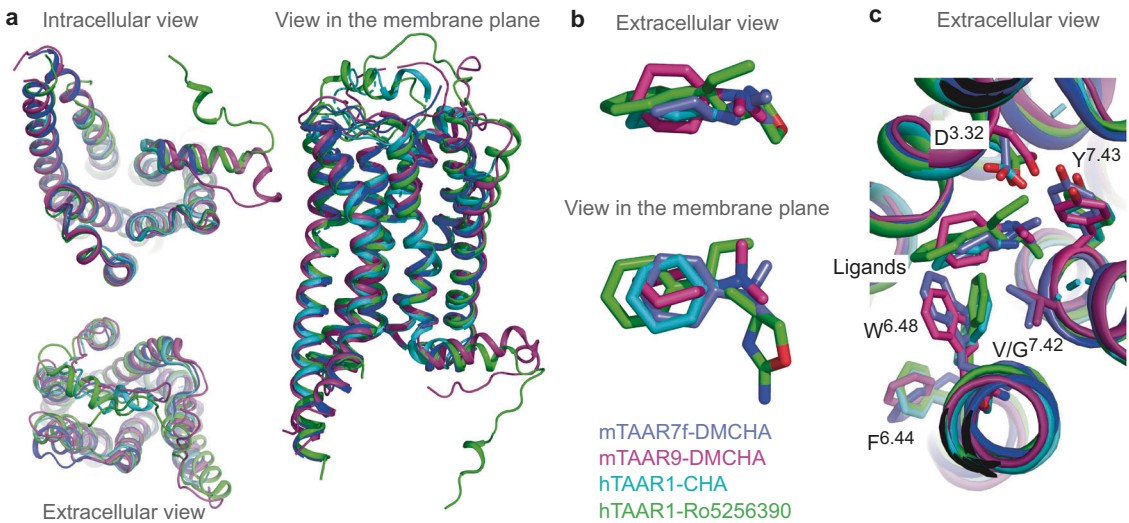

**Fig. 6 | Comparison between structures of mTAAR7f, mTAAR9 and hTAAR1.** **a** Superposition of the agonist-bound G-protein-coupled structures of mTAAR7f (blue), mTAAR9 (magenta; PDB ID 8ITF), hTAAR1 (pale blue; PDB ID 8WCB) and hTAAR1 (green; PDB ID 8UHB). **b** Overlay of ligands derived from receptor alignments in (**a**). **c** Comparison of side chain positions in the receptor alignments.

levels of DMCHA-induced G protein coupling, consistent with their roles in receptor activation (Fig. 3a–e). The mutant D127A[3.32] significantly decreases agonist-induced G protein coupling, and the mutant C131A[3.36] has a similar effect; the structure suggests that C131A[3.36] is important in maintaining the rotamer of Asp127[3.32] for optimal binding to DMCHA. Other mutations (Y132A[3.37], Y289A[6.51]) also reduce agonist-induced signalling, probably by reducing the strength of DMCHA-mTAAR7f interactions.

Full atomistic MD simulations are inadequate to observe the full transition between an inactive state of a GPCR to an active state or vice versa. In the simulations performed here to look at the movements of residues and secondary structure within mTAAR7f in the absence of the G protein and/or DMCHA, we analysed overall trends in the context of deactivation. Five simulations were performed (1 μsec each) either on TAAR7f-DMCHA-mini-G$_s$, TAAR7f-DMCHA or TAAR7f (see 'Methods', Supplementary Fig. 7a–c). In the absence of G protein, the mean GPCR backbone RMSD increased as expected due to the lack of strong stabilisation of the GPCR active conformation through G protein allosteric coupling (Supplementary Fig. 7a)[23,24]. In addition, the simulations show that the OBS increases in volume when ligand and G protein are removed, which is consistent with receptor deactivation. Observation of the activation microswitches in the simulations (Supplementary Fig. 7b, c) also indicated that they all started to move towards inactive state conformations, as assessed by measuring distances between specific pairs of residues; similar results were observed for β$_2$AR (Supplementary Fig. 7c). In contrast, an analogous analysis on OR51E2 showed a distinct series of changes upon removal of the agonist and G protein[9], suggesting that the deactivation process is different from mTAAR7f and β$_2$AR.

## Discussion

During the preparation of the manuscript structures were published of mTAAR9[25] and hTAAR1[26,27]. The amino acid sequence of mTAAR9 (excluding the N-terminus and C-terminus) is very similar to mTAAR7f (71% identity), whilst hTAAR1 shares only 40% sequence identity. However, the overall architecture of all the TAAR structures is highly conserved with overall RMSDs of 1.1 Å (Fig. 6a). The biggest differences are in the extracellular regions and in particular the disposition of ECL2. All of the ligands are in a similar position in the OBS (Fig. 6b) and engage with the conserved Asp[3.32] via the tertiary amine moiety in the ligand. The structures of mTAAR7f and mTAAR9 were both

determined bound to the same ligand, DMCHA. However, the pose of DMCHA in mTAAR9 is different from that in mTAAR7f, with the cyclohexylamine ring rotated by 55° around an axis defined between the tertiary amine and cyclohexylamine ring. The positions of DMCHA in both structures are within the range of poses we observe during MD simulations. Structures of hTAAR1 and mTAAR9 have been determined bound to multiple different ligands which gives interesting insights into how the OBS can accommodate ligands of different sizes and starts to explain ligand specificity[25,27]. The structure of mTAAR9 was also determined coupled to G$_{olf}$ as well as G$_s$; there were no significant differences between the OBS and the pose of ligand binding, suggesting that G$_s$-coupled TAAR structures give valuable insights into the mode of ligand binding despite being coupled to a similar, but non-physiologically relevant G protein[25].

As expected from the comparison of the four active state structures of the TAARs, the position of key residues in receptor activation are in similar positions. For example, Phe[6.44] of the PIF motif occupy the same position in all the structures, although their rotamers vary slightly as would be expected when comparing structures of different sequences (Fig. 6c). In addition, the positions of Trp[6.48] and Val[7.42] in mTAAR7f and mTAAR9 are also very similar. However, in hTAAR1, the Gly residue at position 7.42 allows Trp[6.48] to rotate towards transmembrane helix 7 so that it adopts a position similar to that found in β$_2$AR, which also has a Gly residue at this position. The analysis of the activation mechanism based on mutational analysis broadly agrees with that presented here, namely that the mechanism is most similar to that of β$_2$AR and that Trp[6.48] plays an important role in ligand-induced G protein coupling. However, elucidating the precise mechanism of activation will have to await the structure determination of TAARs in the inactive state.

## Methods

### Expression and purification of the mini-G$_s$ heterotrimer and Nb35

The components of the heterotrimeric G protein (mini-G$_s$ construct 399, β$_1$-subunit, γ$_2$-subunit and Nb35) were expressed and purified as described previously[21,29,30]. In brief, mini-G$_s$ in plasmid pET15b was expressed in bacterial strain BL21-CodonPlus(DE3)-RIL. His-tagged protein was purified via Ni$^{2+}$-affinity chromatography, followed by cleavage of the histidine tag using TEV protease and negative purification on Ni$^{2+}$-NTA to remove the TEV and undigested mini-G$_s$. β$_1$ and

unlipidated (C68S mutation) $\gamma_2$ subunits were co-expressed in High Five (*Trichoplusia ni*) cells (Expression Systems; we did not test for mycoplasma). The protein was purified via Ni$^{2+}$-affinity chromatography followed by anion exchange chromatography. Aggregates in the purified $\beta_1\gamma_2$ complex were removed by size-exclusion chromatography (SEC). The three G protein subunits were mixed and the heterotrimeric G protein isolated by SEC, concentrated, aliquoted and flash-frozen in liquid nitrogen until further use. Nanobody-35 (Nb35) was expressed from plasmid pET26b in the periplasm of *E. coli* strain BL21-CodonPlus(DE3)-RIL, extracted, and purified by Ni$^{2+}$-affinity chromatography, according to previously described methods, followed by ion exchange chromatography[21]. Purified Nb35 was concentrated and flash frozen in liquid nitrogen until further use.

### Cloning and expression of the mTAAR7f
The wild type murine TAAR7f gene (UniProt Q5QD08) was synthesised and cloned into a modified pFastBac1 vector with HA signal sequence, FLAG tag, 10x His-tag and TEV protease cleavage site before the receptor N-terminus and HRV 3C cleavage site followed by eGFP after its C-terminus. Cloning was performed by overlap extension PCR using *Escherichia coli* DH10B cells (Thermo Fischer) and positive clones identified by DNA sequencing. High titre (>3 × 10$^8$ viral particles per ml) recombinant baculovirus was obtained using the Bac-to-Bac expression system (Invitrogen) in Sf9 (*Spodoptera frugiperda*) cells grown in Sf-900 II medium (Thermo Fischer) and its titre was checked with flow cytometry technique using anti-gp64 conjugated antibodies[31]. *Trichoplusia ni* High Five cells (Thermo Fisher Scientific; we did not test for mycoplasma) were grown in suspension in ESF921 media (Expression Systems) and infected at a density of 2–3 million cells per ml using a multiplicity of infection of 7–10. The cells were then collected by centrifugation and resuspended in m7-glycerol+ buffer (20 mM HEPES/KOH pH7.5, 150 mM potassium chloride, 10 mM sodium chloride, 10 mM magnesium chloride, 20% v/v glycerol) supplemented with 1 tablet/50 ml Complete protease inhibitor (Roche), 1 mM PMSF), flash-frozen in liquid nitrogen, and stored for several months at −80 °C until further use.

### Purification of mTAAR7f
Cells were thawed and lysed by two washes in low salt buffer (25 mM Na HEPES pH 7.5, 1 mM EDTA, Complete protease inhibitor, 1 mM PMSF) followed by two washes in high salt buffer (25 mM Na HEPES pH 7.5, 1 mM EDTA, 1 M NaCl, Complete protease inhibitor, 1 mM PMSF), followed by one wash in the low salt buffer. During each round, the pellets were resuspended using an Ultra-Turrax homogeniser and centrifuged (235,000 × *g*, 60 min 4 °C). The pellets were resuspended in m7-glycerol+ buffer supplemented with PMSF and Complete protease inhibitor and flash frozen in liquid nitrogen.

Previously frozen cell membranes containing overexpressed mTAAR7f receptor were thawed and resuspended to a final volume of 160 ml in m7-glycerol+ buffer which was supplemented with 2 mg/ml of iodoacetamide, 6.7 mM N,N-dimethylcyclohexylamine (DMCHA) and EDTA-free complete protease inhibitor tablets (Roche). The mixture was incubated at 4 °C with rotation for 2 h. LMNG/CHS mixture (5/0.5% w/v stock) was added to the final concentration of 1/0.1% LMNG/CHS to solubilise the receptor (4 °C, 1 h) and then centrifuged (430,000 × *g*, 1.5 h, 4 °C). The supernatant was incubated overnight with a 2 ml bed volume of Super Ni-NTA Affinity Resin (Protein Ark) and supplementing with 20 mM imidazole at 4 °C with rotation. All further purification steps were performed at 4 °C. The following day the resin was placed into an empty PD10 gravity column and washed with 10 ml of buffer m7 supplemented with 8 mM ATP, 20 mM imidazole, 6.7 mM DMCHA and 0.01/0.001% LMNG/CHS. The resin was further washed with 15 ml of buffer m7 supplemented with 40 mM imidazole, 6.7 mM DMCHA and 0.01/0.001% LMNG/CHS. The receptor was eluted with elution buffer containing 20 mM HEPES/KOH pH7.5, 150 mM potassium chloride, 10 mM sodium chloride, 10 mM magnesium chloride, 20% v/v glycerol, 300 mM imidazole, 6.7 mM DMCHA and 0.01/0.001% LMNG/CHS. Eluted fractions were pooled and concentrated in a 50 kDa molecular weight cut-off Amicon ultracentrifugal concentrator (Merck) at 2000 × *g* and exchanged into the same buffer used for elution (without imidazole) using a PD10 desalting column. To cleave off the His tag, the protein was incubated overnight with TEV protease in the presence of 0.5 mM DTT.

The purity of mTAAR7f was then improved significantly using a reverse Ni-NTA purification step of the TEV cleaved mixture by incubating it with rotation for 2 h with 0.5 ml bed volume of Super Ni-NTA Affinity Resin (Protein Ark) supplemented with 10 mM imidazole. eGFP was then cleaved off by incubation with HRV 3C protease in the presence of 0.5 mM DTT. The cleaved receptor was separated from eGFP by SEC on a Superdex 200 10/300 GL column (GE Healthcare) pre-equilibrated with m7-glycerol+ buffer containing 6.7 mM DMCHA. Peak fractions were pooled and concentrated using a 50 kDa molecular weight cut-off Amicon ultracentrifugal concentrator (Merck).

### mTAAR7f–miniGs399–$\beta_1$–$\gamma_2$–Nb35 complex assembly
Purified and concentrated mTAAR7f was mixed with about 10x molar excess of both the heterotrimeric G protein and Nb35 and 0.5 U apyrase (NEB) and incubated overnight. The following morning unbound G protein and Nb35 were separated from the complex by SEC on a Superdex 200 10/300 GL column (GE Healthcare) pre-equilibrated with the buffer containing 20 mM HEPES/KOH pH7.5, 150 mM KCl, 10 mM NaCl, 10 mM MgCl$_2$, 6.7 mM DMCHA, and 0.01/0.001% LMNG/CHS. Fractions corresponding to the size of the complex were pooled, and concentrated in a 100 kDa cut-off concentrator.

### Grid preparation of the complex and data collection
Grids for cryo-EM (UltrAuFoil 1.2/1.3) were prepared by applying 3 µl sample concentrated to 0.9 mg/ml on a glow-discharged grid (2 min in Ar-Oxy 9-1 plasma chamber, at Forward Power of 38 W, Reflected Power of 2 W; Fischione). The excess sample was removed by blotting for 3 s before plunge-freezing in liquid ethane (cooled to −181 °C) using a FEI Vitrobot Mark IV maintained at 100% relative humidity and 4 °C. Data were collected in-house from a single grid on the FEI Titan Krios microscope at 300 kV equipped with a Falcon 4 detector in counting mode. A total of 12,273 movies were collected in one session with a fluence of 55 $e^-/Å^2$ at ×96,000 magnification (0.824 Å/pixel). The gain reference file was provided by the facility and used unmodified.

### Cryo-EM data processing
A total of 12,273 movies in .EER format were converted into .tiff format with relion_convert_to_tiff utility[32,33], grouping the frames to get the dose per frame of 1.38 $e^-/Å^2$. The resulting movie stack was imported into CryoSparc v4.1.1+patch230110[34] and the processing was performed there unless specified otherwise (Supplementary Fig. 3). Overall, drift, beam-induced motion and dose weighting were corrected with Patch Motion Corr. CTF fitting estimation were performed using Patch CTF estimation. The exposures were manually curated: the only images kept had an estimated CTF resolution of <5 Å, motion distance <200 pixels and no obvious outliers in terms of estimated relative ice thickness. This yielded a stack of 11,157 movies. Auto-picking was performed with Gaussian circular and elliptical blobs as templates with inner and outer diameters 80 and 160 Å respectively and 0.5 diameters as a minimum separation distance. Particle picks were curated to remove obvious junk peaks (e.g. the ones outside of foil holes or on contaminants) and then extracted with the box size of 307 Å and down-sampled to 1.648 Å/pixel. The particles were subjected to five rounds of 2D classification and the clean particle stack was re-extracted at 0.824 Å/pixel.

Ab initio reconstruction for the mTAAR7f-G protein complex was made using 198,414 particles belonging to clean 2D classes with

different orientations. Hetero refinement was performed with the ab initio reconstruction of the receptor complex and three noise classes as input. The good class from hetero refinement was subjected to one round of non-uniform refinement resulting in an initial 3D volume.

The curated particle image coordinate data was exported from cryoSPARC using pyem v0.5[35]. Beam-induced motion correction and dose-weighting were repeated using RELION's implementation of motion correction with a $5 \times 5$ patch array. Particle images were then re-extracted from the averaged micrographs and realigned to the consensus map through non-uniform refinement. Coordinates and transformations were exported with pyem for Bayesian polishing in RELION, maintaining image dimensions of the shiny particle stack.

Per-hole beam image-shift exposure groups were identified with EPU_group_AFIS[36]. Particle images were assigned to exposure groups and refined in cryoSPARC using non-uniform refinement[37], iterated with particle defocus refinement and higher order CTF refinement (beam tilt and trefoil parameters)[38], to an estimated global resolution of 2.92 Å (gold-standard FSC = 0.143; Supplementary Fig. 2e, f).

To help receptor modelling, especially its most flexible regions (helices 1 and 2) focused refinement was performed using a mask on the receptor, which visually improved the map quality. This refinement centred on a mask of the receptor region, using pose/shift Gaussian priors and 3°/3 Å standard deviations, yielded a 3.2 Å focused map. Local resolution estimation was performed using cryoSPARC's adaptive window implementation. The nominal local resolution for the receptor was improved from 3.5 Å in the overall consensus map to 3.2 Å in the focused map. Local sharpening was performed with LocalDeblur[39] using half-maps and estimated local resolution maps.

## Model building and refinement
Initial models of heterotrimeric mini-$G_s$ and Nb35 were sourced from PDB 7T9I. A de novo model of TAAR7f was generated from the focused map and protein sequence using ModelAngelo[40]. Overall, this initial model agreed with the map except for poorly resolved regions which were added and iteratively modelled afterwards. Chemical restraints for N,N-dimethylcyclohexylamine were generated using AceDRG[41] and manually fitted into the density. Manual rebuilding was performed in COOT[42] and ISOLDE[43] (in ChimeraX[44]) and further refined against the locally-sharpened consensus map using Refmac version 5.8.0419[45,46]. 3D variability analysis[47] was performed on the consensus map using the refinement mask and at a filter resolution of 4 Å with a high-pass prior of 20 Å.

## Molecular dynamics simulations
Three distinct molecular dynamics simulations were performed to investigate TAAR7f behaviour under different conditions: (1) an apo state simulation, excluding both ligand and G protein; (2) a ligand-bound simulation, incorporating the ligand but devoid of the G protein; (3) a G protein-bound simulation, including the ligand and mini-$G_s$, but omitting the β-subunit and γ-subunit of the heterotrimeric G protein and Nb35. The starting point for these simulations was the cryo-EM structure of the DMCHA-bound TAAR7f-$G_s$ complex. TAAR7f in each simulation was encapsulated within a layer of cholesteryl hemisuccinate, employing the same methodology as that used in our prior OR study[9]. Utilising the membrane builder module[48] from CHARMM-GUI[49], the simulation system was assembled by embedding the complexes into a POPC bilayer with X-Y dimensions of 85 Å–85 Å for apo and ligand-bound systems, and 125 Å–125 Å for the G protein-bound system. The resulting system was then immersed into a TIP3 water box providing a 10 Å margin along the Z-axis from the protein surface (Z dimension ~125 Å), followed by neutralisation using 0.15 M NaCl. The simulations systems were parameterized by CHARMM36m force field[50], and all simulations were performed by GROMCS-2022 version[51].

An additional MD simulation was established to scrutinise the ligand association process. This involved positioning apo state TAAR7f (no G protein involved) within a grid box with 8 * 8 * 8 units of DMCHA, with a grid spacing (distance between any two adjacent DMCHA residues) of 12 Å. This complex was configured into a simulation box by adopting the same procedure previously outlined, after the removal of any DMCHA molecules found to overlap with the POPC bilayer.

The simulation systems were progressively heated from 0 K to 310 K using a constant volume-constant temperature (NVT) ensemble and a Nosé-Hoover thermostat[52]. This was followed by a 30 ns equilibration protocol implemented using a constant pressure-constant temperature (NPT) ensemble. Throughout both heating and equilibration phases, harmonic positional restraints were applied to proteins, the ligand, and the heavy atoms of the head group of the cholesteryl hemisuccinate and POPC lipids. The system was initiated with a positional restraint force constant of 10 kcal/mol-Å², which was upheld for a duration of 5 ns. This was succeeded by a decrease in the constant to 5 kcal/mol-Å², which was maintained throughout the next 5 ns. The constant was then methodically reduced to 0 kcal/mol-Å² at a decrement rate of 1 kcal/mol-Å² for each successive 5 ns span. The concluding phase of the equilibration process was carried out with a restraint constant of 0 kcal/mol-Å² over a 10 ns period. Pressure control was facilitated by the Parrinello-Rahman method[53], and the simulation system was harmonised with a 1 bar pressure bath. The concluding snapshot from the equilibration stage was chosen as the commencement conformation for five unrestrained NPT simulation runs, each with distinct random seeds. Each of these runs spanned 1000 ns at a temperature of 310 K. In the case of the ligand association simulations, the duration of each run was extended to 2200 ns. In all the simulations, the LINCS algorithm was utilised for all water bonds and angles, with a time step of 2 fs for integration. Non-bond interactions were subjected to a cut-off of 12 Å, and the particle mesh Ewald method was used to handle long-range Lennard-Jones interactions[54]. Molecular dynamics snapshots were saved at intervals of every 20 ps.

To ascertain the flexibility of the ligands, we conducted ligand clustering analysis utilising the cluster-analysis-using-VMD-TCL script (https://github.com/anjibabuIITK/CLUSTER-ANALYSIS-USING-VMD-TCL). The merged production trajectories from both the ligand-bound and G protein-bound simulations were initially aligned using the backbone atoms of TAAR7f. Following this, a cut-off value of 1.5 Å was utilised to group ligand conformations based on the root-mean-square deviation (RMSD) of their heavy atoms (Fig. 1d).

To scrutinise the TAAR7f residues establishing stable contacts with the DMCHA ligand, we implemented contact frequency analysis utilising the 'get_contact' script (https://getcontacts.github.io/). This analysis was performed on the combined production runs from both the ligand-bound and G protein-bound simulations. TAAR7f and the DMCHA ligand were designated as selection 1 and selection 2 respectively. All categories of contacts were considered in the analysis, and the default parameters were employed to evaluate the formation of contacts (Fig. 2e).

Using the MDAnalysis module[55], we calculated the average ligand RMSD for each of the five individual production runs in both the ligand-bound and G protein-bound simulations. The calculation was performed on the ligand's heavy atoms, subsequent to aligning the backbone atoms of TAAR7f. This resulted in five average RMSD values for each simulation. We compared these five values, presenting their average and standard deviation in a bar graph (Fig. 1d). The graph also includes $p$ values derived from a $t$-test, offering a statistical comparison between the two simulations.

The calculation of GPCR RMSD was conducted in a similar manner to the ligand RMSD, but utilised the backbone atoms of TAAR7f. No G protein was involved in this calculation. The average TAAR7f RMSD was determined for each individual production run for the Apo, ligand-bound, and G protein-bound simulations, and the results were plotted in a bar graph with accompanying p-values, as depicted in (Supplementary Fig. 7a).

The calculation of ligand binding site volume was conducted using the Maestro SiteMap module (Schrödinger Release 2023-2: SiteMap, Schrödinger, LLC, New York, NY, 2021.). For each production run across all simulations, frames were extracted at the end of both 500 ns and 1000 ns of simulations, resulting in 10 frames per simulation. In the case of the ligand-bound and G protein-bound simulations, the binding pocket was defined by centring on the ligand, and a 6 Å cut-off was applied to establish the binding region. We employed the 'standard grid' with a 'more restrictive definition of hydrophobicity', and the site was truncated 4 Å from the nearest site point. In the Apo simulation, the ligand was first docked into the vacant binding site for the 10 frames, and then the same procedure was followed to calculate binding site volume. The ten volume values from each simulation were plotted as a bar graph, with associated $p$ values (Supplementary Fig. 7a).

The Chi2 dihedral angle of W286 was determined using the MDAnalysis module with the dihedral angle sequence: CA-CB-CG-CD2, designated as the Chi2 angle. Subsequently, these dihedral angles were represented in a histogram constructed using the numpy.hist function, with a bin width of 50 (Supplementary Fig. 7c).

Using MDAnalysis, we determined microswitch distances by calculating the shortest separations between specified atom pairs in various configurations. These include the D127-Y316 distance (between OD1/OD2 atoms of D127 and the OH atom of Y316), the sodium binding site distance (OD1/OD2 atoms of D93 and the OG atom of S134), the NPxxY motif distance (OD1/ND2 atoms of N322 and the OH atom of Y326), and the YY motif distance (OH atoms of Y326 and Y232). This analysis was performed on residues sharing the same Ballesteros–Weinstein numbering in the $\beta_2$AR simulation retrieved from GPCRmd.org (apo state, ID 116; active state, ID 117).

We executed ligand clustering analysis on the four ligand association trajectories using the TICC module from the get_contact script. In these trajectories, different DMCHA residues associate with TAAR7f. We limited the association trajectory to include only TAAR7f and the specific DMCHA residue in association with TAAR7f, excluding all other non-bound DMCHAs. Initially, we computed the contact frequency between the associated DMCHA and TAAR7f following the procedure described earlier. We then clustered the frames using the TICC module, arbitrarily setting the cluster count to five. Upon comparison of the representative conformations from each cluster across the four association trajectories, we identified three stable states, which are further elaborated in the results section. Regarding distance measurements, the ligand-D296/D296 distance was quantified as the minimum distance between the N atom of DMCHA and the OD1/OD2 atoms of aspartate. On the other hand, the ligand-Y308/W286 distance was determined by measuring the centre of mass distance between the cyclohexane ring of DMCHA and the heavy atoms of Y308/W86 (Supplementary Fig. 5).

### Signalling assays

Full length mTAAR7f was cloned into plasmid pcDNA4/TO, encoding a signal peptide, a twin-strep tag, a SNAP-tag at the N-terminus of mTAAR7Ff and nanoluc at the C-terminus. Stable cell lines were made in HEK293 T-ReX cell line (Thermo Fisher) by adding Zeocin selection agent to kill non transfected cells. Then, 100 µL aliquots of the stable cell line were seeded in 12 wells of a 96 well white plate with clear flat bottom (25,000 cells per well). Cell lines expressing wild type mTAAR7F and different mutants were transiently transfected with a plasmid expressing NES-Venus-mGs[56] (kind gift of Nevin Lambert's lab) and cultured in T75 flasks in DMEM, supplemented with 10% FCS, at 37 °C in a humidified incubator with 5% CO$_2$. After reaching 80%–90% confluency, cells were induced for 48 h with 1 µL/mL tetracycline. The growth media was aspirated, and cells were washed once with 90 µL of assay buffer (HBSS + 0.5% BSA + 0.5 mM HEPES pH 7.4 + 0.01% ascorbic acid), which was warmed to 37 °C in a water bath. The nanoluc

substrate furimazine was added to assay buffer (8 µM final concentration), and 90 µL of this nanoluc substrate containing assay buffer was added to each well. To improve signal level, a white sticker was attached to the bottom and the plates were read in a PHERAstar FSX Microplate reader using an optic module Lum 550-LP 450-80 for 10 min before adding any compound. Finally, 10 µL of compound dilutions were added to the wells and the plate reading was continued for a further 30 min. The maximum possible final concentration of DMCHA that was reproducibly attainable was 1 mM due to its relative insolubility in aqueous buffers. The data were analysed using GraphPad Prism 9 using standard concentration-response models defined in the software.

Cell surface receptor expression was determined by labelling the SNAP tag at the N-terminus of the receptor with Alexa Fluor 488 and SNAP-Cell® 647-SiR. The stable cell lines (100 µL) were seeded in 12 wells of a 96 well white plate with clear flat bottom (25,000 cells per well). After reaching 80–90% confluency, 1 µL/mL tetracycline (10 mg/mL stock) was added to induce the cells for expression. After a minimum of 48 h incubation of plates at 37 °C, in a humidified atmosphere with 5% CO$_2$, the non-permeable dye Alexa Fluor 488 was added to a well (1 µM final concentration) and incubated for 15 min at 37 °C, in a humidified atmosphere with 5% CO$_2$. Then a permeable dye, SNAP-Cell® 647-SiR, was added to the same well and the plates were incubated again, this time for 30 min. In a second well, only Alexa Fluor 488, and to a third well, only SNAP-Cell® 647-SiR were added as controls. After incubation, the cell in each well were detached by resuspension and transferred to a microcentrifuge tube, then centrifuged for 5 min at 16,000 × $g$ to pellet the cells. After removal of the supernatant, the cells were lysed with 1x LDS buffer in assay buffer (HBSS + 0.5% BSA + 0.5 mM HEPES pH 7.4 + 0.01% ascorbic acid). Centrifuged again at 16,000 × $g$ (10 min) and the supernatant loaded on NuPAGE Bis-Tris SDS-PAGE gels and resolved (90 V for 45 min). Gels were imaged with a Typhoon using CY2 (473/520) and CY5 (635/670) channels. Finally, gels were stained overnight in SYPRO reagent (1:5000 dilution of stock in 7.5% (v/v) acetic acid) before being imaged again. Gel images were analysed with FIJI (ImageJ) software.

### Reporting summary

Further information on research design is available in the Nature Portfolio Reporting Summary linked to this article.

## Data availability

The source data underlying Fig. 3a–d are provided as a Source Data file. The cryo-EM maps have been deposited in the Electron Microscopy Data Bank (EMDB) under accession code EMD-17756 (density maps of mTAAR7f). The atomic coordinates have been deposited in the Protein Data Bank (PDB) under accession code PDB 8PM2 (mTAAR7f model). The cryo-EM images have been deposited in the Electron Microscopy Public Image Archive (EMPIAR) under accession code EMPIAR-12101. There are no restrictions on data availability. Previously published PDB codes are available as follows: 2RH1, 3SN6, 4LDO, 7E32, 7JJO, 7T9I, 7XT8, 8F76, 8ITF, 8WCB, 8UHB. Source data are provided with this paper.

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

## Acknowledgements

We thank R. Axel, S. Liberles, V. Velazhahan, K. Yamashita, G. Murshudov, B. Ahsan, and T. Warne for helpful discussions. We acknowledge the MRC Laboratory of Molecular Biology Electron Microscopy Facility for access and support of electron microscopy sample preparation and data collection, the LMB scientific computing for technical support and the Flow Cytometry facility for support in cell analysis. We also thank S. Liberles for supplying original clones of receptors. The work in C.G.T.'s laboratory was supported by core funding from the Medical Research Council [MRC U105197215] and by a grant from Sosei Heptares. Work in the lab of F.M. was funded by the National Institutes for Health (USA) grant GM132120. S.N.W.'s lab was funded by a Sir Henry Dale fellowship from the Wellcome Trust and the Royal Society London (Grant Number 101234/Z/13/Z) and by the Isaac Newton Trust Cambridge (Grant Number 15.40(a)). The work in N.V.'s lab was funded by grants from the National Institutes of Health (2R01-GM117923, R01-DC020353 and R01-DC021585). A.N.K. is supported by the BBSRC Doctoral Training Programme at the University of Nottingham. For the purpose of open access, the MRC Laboratory of Molecular Biology has applied a CC BY public copyright licence to any Author Accepted Manuscript version arising.

## Author contributions

A.G. optimised the constructs, performed receptor expression, purification, preparation of cryo-EM grids, cryo-EM data collection, data processing, structure determination and model building. F.H. developed the initial expression and purification strategy. P.C.E. performed the expression and purification of G protein. Q.C. cloned receptor variants for functional studies. Y.L. advised on receptor expression, purification and data collection, supervised data processing, structure determination and performed model building. E.M. and N.M. performed the molecular dynamics simulations, and N.M. and N.V. did the analysis of the MD simulation trajectories. A.N.K. and E.K. performed G protein-coupling assays, and A.N.K., E.K. and D.V. did the analysis of the data. J.K. and F.M. contributed significantly at the initial stages of receptor screening and selection. A.G. and C.G.T. carried out structure analysis and manuscript preparation. C.G.T. and S.N.W. managed the overall project. The manuscript was written by A.G. and C.G.T., and included contributions from all the authors.

## Competing interests

The authors declare the following competing interests: C.G.T. is a shareholder, consultant and member of the Scientific Advisory Board of Sosei Heptares. Unique materials described in this paper are freely available upon reasonable request. The remaining authors declare no competing interests.
