## [Peer Review File · Nature Communications]

Molecular recognition of an odorant by the murine trace amine-associated receptor TAAR7fREVIEWER COMMENTS

Reviewer #1 (Remarks to the Author):

The manuscript by Tate et al provide important insights for odor sensation by olfactory receptor TAAR7f, which may have great impacts in research field of mammal sensation and GPCR biology. Importantly, the odor sensation is the basic senses that allows animals to detect a variety of chemical stimuli or odorants from the environment. As one of the main olfactory receptor families, the trace amine-associated receptor (TAAR) family developed specialized mechanisms to detect amines in animal bodily fluids and elicit innate behaviors such as attraction of female to male, seeking food, escaping from predators. Therefore, research on the functions and structures of TAARs is highly significant. In current study, Gusach et al determined the cryo-EM structure of a murine receptor, mTAAR7f, coupled to the heterotrimeric G protein Gs and bound to the odorant DMCH to an overall resolution of 2.9Å. DMCH is bound in a hydrophobic orthosteric binding site primarily through van der Waals interactions and a strong charge-charge interaction between the tertiary amine of the ligand and an aspartic acid residue. The structure, in combination with mutagenesis data and molecular dynamics simulations suggests that the activation of the receptor follows a similar pathway to that of the adrenoceptors, with the significant difference that DMCH interacts directly with one of the main activation microswitch residues. However, the ambiguous ligand map density and poor data quality of pharmacological characterization jeopardized the overall explanation of their discoveries, and even may lead to wrong conclusions. In particular, they didn't give enough credit to the recently published mTAAR9 work, which are of high quality and value in both number of structure solved, data quality of both cryo-EM data and pharmacological analysis, as well as much earlier timeline. Therefore, I have summarized important concerns that should be addressed so that the current study (mTAAR7f) qualifies the journal's scientific level and adds their important impacts in the GPCR research field.

Major concerns

1. The first TAAR family structure has already been reported in Nature (Nature. 2023 Jun;618(7963):193-200). In that study, the authors describing the first structure of TAAR family, TAAR9, and its recognition of natural odorant chemicals, including a polyamine, spermidine (SPE) and a monoamine, β -phenethylamine (PEA), and the first report elucidating the structural mechanism by which an olfactory G protein (Golf) couples to an olfactory receptor. They discovered that mTAAR9 recognizes amine odorants within a deep and tightly bound pocket. A distinctive disulfide bond, conserved throughout the TAAR family, is crucial for activating TAAR family members in response to odorants. This research (Nature. 2023 Jun;618(7963):193-200) identify motifs of TAARs critical for recognition of both primary and other minor amino groups. Moreover, the motifs involved in specific odor chemical sensation triggered by a selective subset of TAAR members are elucidated, providing preliminary insights into combinatorial odor coding mechanisms. These studies, both in terms of structure and pharmacology research, serve as outstanding templates for comprehending the mechanisms of TAARs. However, the current manuscript of mTAAR7c only provide negative comments to the mentioned seminal work (Nature. 2023 Jun;618(7963):193-200), which is unfair. The mTAAR9 work (Nature. 2023 Jun;618(7963):193-200) is published earlier than the submission of current mTAAR7f work. I therefore suggested that the author in current mTAAR7c should compared their work with mTAAR9 work in a rational way and provide positive comments. Especially, the authors should primarily use TAAR9 as a template for structural comparison and description, rather than OR51E2.
2. There are multiple concerns regarding to the cryo-EM maps and models provided. Positioning of the ligand DMCH into the cryo-EM densities were ambiguous. In current EM map, the CMCH could be fit with both 180 and 90 degree turns. The authors should consider deposition of multiple models with different orientations or run computational simulation for these binding poses, supplemented with mutational analysis to verify the conformation of the compound.
3. For the DMCH binding pockets, I could not find any descriptions for Fig2b in the manuscript, I don't

know if it was caused by negligence or other reasons. Especially for DMCH binding, which is key conclusion in the manuscript, should be supplemented with MD and Radio-ligand binding experiments for examining their conclusions.

4. It should be emphasized that in order to provide scientific conclusion in the manuscript, the pharmacological or cellular activity measurements of the mutants are essential, including characterizations of the binding site, G protein interface, and activated motif. Unfortunately, we only saw data for 2F and the corresponding EDF 9, and this data was of extreme poor quality, including unusual large error bar. Moreover, the baseline value of each mutation was different, suggesting unequal expression of the receptors. There is also no data on the expression of receptor mutants, which is essential to ensure accurate conclusion. In addition to optimizing BRET experiments, I will suggest measurements of the cAMP value should be considered for functional readout of the mutants.

5. In line 164-172, it was said that "Previous structure-activity relationship (SAR) data for mTAAR7f suggests that ligand binding is highly dependent on the ligand shape and the length of the hydrophobic chain, with aliphatic chains containing less than six carbon atoms being unable to activate the receptor". I think the author should provide this SAR data or use a diagram to provide readers with a more intuitive view of the limitation of a certain residue on the ligand volume bound to mTAAR7f. The series of conclusions brought about by the Y132C mutation given later should be supported by corresponding data. I believe that a biochemical mutation data is necessary.

6. There are same issues regarding to the G protein interface. The author compared the G protein interface of mTAAR7f with other receptors and provided a barcode (fig 3b), but the corresponding mutation data is missing. It is necessary to provide mutation activity data for the corresponding site.

7. According to the map and model provided by the author, it is not difficult to find that many side chains of residues are not placed correctly, such as R107, V209, W213, etc. Additionally, there are many electron densities that are not well explained. Therefore, it is necessary for the author to redefine the model and correct Ramachandran plots and geometric issues.

8. To ensure the accuracy of the model, it is recommended that the author add a figure showing the density of the amino acids involved in the binding pocket and the amino acids of the G protein interface in the manuscript.

9. Unfortunately, I did not see a table summarizing structural and refinement statistics of the new structure in the manuscript, this is the most basic raw data for for the Cryo-EM structure, which also worries me whether some authors of this manuscript are not from a professional structure lab (Although we know Dr. Tate is very good structural biologist).

Minor concerns

1. The reference to Fig2d in line 129 is incorrect and does not match the text. Please correct it.
2. The vitality chart displayed in Extended Data Figure 9 is not satisfactory. The error bar in the figure is too large. I think authors should repeat these experiments to make more accurate measurements.
3. The sequence comparisons in Fig1a are all based on human-derived GPCRs, whereas the main subject of this paper is the murine TAAR7f. I believe that the author need to supply this figure with members of murine TAARs, including mTAAR7f.
4. For molecular dynamics simulations such as Fig1D, Fig2E, etc., the RMSD over time needs to be represented in the form of a line graph to obtain more detailed information.
5. Please note that many scale bar are missing for supplementary figures.

Reviewer #2 (Remarks to the Author):

This manuscript by Gusach et al reports the structure of trace amine receptor TAAR7f bound with one of its preferred ligands DMCH and coupled to Gs, as derived from cryoEM. Overall, the work appears sound and largely corroborates very recently published work on a similar receptor, TAAR9 from another group. The publication of the structure of TAAR receptors is timely given the also recent publication of the first structure of a canonical odorant receptor OR51E2. I have a few minor comments and clarifications with the text.

While a description of TAAR structure is clearly a landmark (given the publication of the prior paper in Nature), the conceptual advance provided is perhaps somewhat incremental. This class A GPCR shares many of the same properties as the well-studied adrenergic receptors. That being said, I would think this study should be published on the heels of the previously published work on TAAR9 so that the community can compare and contrast the results.

A few minor comments

1. Figure 1a: I don't think this figure clearly demonstrates the point that the text puts forward. It is difficult to see the distance between the TAARs and the adrenergic receptors. Most of the tree focuses on irrelevant receptors (which are also boldly colored). Perhaps it would help to make an unrooted tree and color the receptors/branches to better highlight what the readers should pay attention to.
2. Extended Data Figure 9: These are primary data that are important for the main point of the manuscript and thus should be included in the main figures. Simply reporting pEC50 and Emax is not sufficient. Also, the authors should explain the reason for the differences in baseline offset for the BRET measurement across the different mutations.
3. The title references the fact that the TAAR7f ligand used is "aversive". However, the text says that mice show "either attractive/neutral or aversive behaviour when exposed to TAAR7f ligands, such as amines found in urine (ref 13)"

Detailed comments

Line 63: "...the TAAR studied here,..." Please clarify whether all the TAARs are more similar to adrenergic receptors as compared with ORs, or is it just TAAR7f that is more similar to adrenergic receptors.

Line 66: The authors should cite original articles for TAAR behaviors (PMID: 23177478; PMID: 23624375; PMID: 30158871)

Line 68: "TAAR7f is a well-studied murine homolog..." This line could be interpreted as TAAR7f being the closest mouse homolog of human TAAR9, which it is not. Also it is not clear to which receptor the phrase "has well characterized agonists" refers.

Line 70: "elicit" should say "exhibit"

Line 103: "Due to the lack of..." The authors should clarify what they mean by "the ligand was placed according to its known functional properties". On the surface, this sounds like circular reasoning (the authors placed the ligand where they thought it should go).

Line 281: The authors should avoid the term "worse" as it is imprecise and potentially inflammatory.

Line 281: "...but the position is in line..." The authors should clarify which position—the one they observe or the one previously published.

Reviewer #3 (Remarks to the Author):

There are two main families of G protein-coupled receptors that detect odours in mammals, the odorant receptors (ORs) and the trace amine-associated receptors (TAARs). In spite of their importance for olfactory signal detection and transduction, there are presently only the high-resolution 3D structure of one OR (OR51E2) and one TAAR (mTAAR9) available. Using cryo-EM, the present manuscript delivers the structure of another trace amine-associated receptor, the mTAAR7f, coupled to the heterotrimeric G protein Gs and bound to the odorant DMCH. Combined with mutagenesis data

and molecular dynamics simulations an activation mechanism is proposed which shows similarities but also differences to the known activation pathway of beta-adrenoceptors. Although the structure of mTAAR7f and mTAAR9 are similar, the pose of DMCH bound to mTAAR7f is different to that found in mTAAR9. Also, the molecular dynamics (MD) simulations presented in the present manuscript deliver important information about contact sites and frequency of contacts between the ligand DMCH and specific amino acid residues in the ligand binding pocket of mTAAR7f. Similar information are provided for the contact sites between the mTAAR7f and its Gs protein. Such data are extremely important for evaluating the kinetic stability of the corresponding interactions which are missing in the published article about the mTAAR9 structure (ref 22 cited in the present manuscript). Taking into consideration of the rare structural data available for receptors involved in olfaction, the results presented in this manuscript are of utmost importance for elucidating the function of TAAR receptors and the signal transduction mediated by GPCRs in general. I recommend publication of the manuscript in Nature Communications without modification.

Reviewer #1 (Remarks to the Author):

Major concerns

1. The first TAAR family structure has already been reported in *Nature* (*Nature*. 2023 Jun;618(7963):193-200). In that study, the authors describing the first structure of TAAR family, TAAR9, and its recognition of natural odorant chemicals, including a polyamine, spermidine (SPE) and a monoamine, β -phenethylamine (PEA), and the first report elucidating the structural mechanism by which an olfactory G protein (Golf) couples to an olfactory receptor. They discovered that mTAAR9 recognizes amine odorants within a deep and tightly bound pocket. A distinctive disulfide bond, conserved throughout the TAAR family, is crucial for activating TAAR family members in response to odorants. This research (*Nature*. 2023 Jun;618(7963):193-200) identify motifs of TAARs critical for recognition of both primary and other minor amino groups. Moreover, the motifs involved in specific odor chemical sensation triggered by a selective subset of TAAR members are elucidated, providing preliminary insights into combinatorial odor coding mechanisms. These studies, both in terms of structure and pharmacology research, serve as outstanding templates for comprehending the mechanisms of TAARs. However, the current manuscript of mTAAR7c only provide negative comments to the mentioned seminal work (*Nature*. 2023 Jun;618(7963):193-200), which is unfair. The mTAAR9 work (*Nature*. 2023 Jun;618(7963):193-200) is published earlier than the submission of current mTAAR7f work. I therefore suggested that the author in current mTAAR7c should compared their work with mTAAR9 work in a rational way and provide positive comments. Especially, the authors should primarily use TAAR9 as a template for structural comparison and description, rather than OR51E2.

We are well aware of the *Nature* paper describing the structures of mTAAR9 bound to different ligands. As the reviewer rightly points out, it is indeed the first published paper describing a TAAR receptor and contains extensive data on how ligands of different structure bind to the same receptor. It also provides the first structure of a TAAR receptor bound to G_{olf} protein and compares it to G_s. However, our structural work was completed before this paper was published and our manuscript is an independent view on the structure of TAARs. Our background knowledge from our experiences is also obviously different from the authors of the *Nature* paper, so we look at things in different ways and give a different perspective. We stand by our decision to use OR51E2 for the comparison of our structure, because this is the most interesting comparison from the perspective of evolution and understanding the differences in activation mechanism. We provided a comparison with mTAAR9 at the end of the manuscript, which is essentially identical to that of mTAAR7f. Our manuscript is not a review and therefore the amount of space apportioned to discussing mTAAR9 is appropriate.

We did not provide 'negative comments' on the mTAAR9 structure, only the truth. The fact that the DMCH ring is modelled differently in mTAAR9 and mTAAR7f cannot be denied and, as we point out, they are both within the range of poses seen in MD simulations. An objective analysis of the ligand density in both structures demonstrates that the density is far weaker in mTAAR9 compared to mTAAR7f. However, at the suggestion of Reviewer 2, this text has been re-written to provide more precision (lines 293-323).

2. There are multiple concerns regarding to the cryo-EM maps and models provided. Positioning of the ligand DMCH into the cryo-EM densities were ambiguous. In current EM map, the CMCH could be fit with both 180 and 90 degree turns. The authors should consider deposition of multiple models with different orientations or run computational simulation for these binding poses, supplemented with mutational analysis to verify the conformation of the compound.

The quality of the cryo-EM maps are as expected for a 2.9 Å resolution model of the complex and is of the same or better quality compared to the other 10 cryo-EM structures we have published of GPCR complexes. We are confident that our model is a good representation of the density, although we have obviously not been able to model flexible loops.

The reviewer seems to be particularly concerned about the modelling of the ligand DMCH and we do not understand why. There is clear density for the ligand (without having to resort to EMDepEnhancer) and as it is a flattened oval, there is convincing evidence for the actual pose modelled. Rotating the ligand through 90° would result in a worse fit to the density. We are also surprised by the reviewer suggesting to 'run computational simulation for these binding poses, supplemented with mutational analysis to verify the conformation of the compound'. Firstly, the simulations had already been performed and were presented in the submitted manuscript (Figure 1). Secondly, mutational analysis would be highly unlikely to be able to discriminate between poses of DMCH that differed by a 90° rotation of the cyclohexylamine ring. This is because the ligand is highly dynamic in the binding pocket (as shown by our MD simulations) and would be expected to make multiple transient interactions to residues throughout the OBS, as our data show. We therefore have not performed any further mutational analysis beyond what we have already presented.

3. For the DMCH binding pockets, I could not find any descriptions for Fig2b in the manuscript, I don't know if it was caused by negligence or other reasons. Especially for DMCH binding, which is key conclusion in the manuscript, should be supplemented with MD and Radio-ligand binding experiments for examining their conclusions.

Fig 2b is fully described in the figure legend. It was not referenced in the text, and this has now been inserted.

We do not understand the reference made by the reviewer stating that our data 'should be supplemented with MD'. As described in our manuscript, we performed extensive MD simulations to describe the motions of the ligand and also how it enters the orthosteric binding pocket and activates the receptor (old manuscript numbering: Fig. 1d, Fig 2e, Extended Data Fig. 6ab, Extended Data Fig. 7a,b, Extended Data Fig. 8; Lines 107, 134-141, 160-163, 213-218, 248-249, 624-719)

Radioligand binding assays are exceedingly useful for studying GPCRs and CGT has over 30 years of experience in using them in both transporters and GPCRs. Regrettably, DMCH is a highly hydrophobic ligand with an EC₅₀ of about 400 nM for TAAR7f (Ferrero et al. 2012), making radioligand binding assays exceedingly challenging due to the fast off-rate and high background. In the odorant receptor field, there are no reports we know of where radioligand binding assays have been successfully performed on native or near-native ligands. There have been reports of radioligand binding assays performed on TAAR1, but the ligand used is a synthetic ligand with a pK_i and EC₅₀ better than 10 nM. No radioligand binding studies were performed in the paper describing the structure of TAAR9.

4. It should be emphasized that in order to provide scientific conclusion in the manuscript, the pharmacological or cellular activity measurements of the mutants are essential, including characterizations of the binding site, G protein interface, and activated motif. Unfortunately, we only saw data for 2F and the corresponding EDF 9, and this data was of extreme poor quality, including unusual large error bar. Moreover, the baseline value of each mutation was different, suggesting unequal expression of the receptors. There is also no data on the expression of receptor mutants, which is essential to ensure accurate conclusion. In addition to optimizing BRET experiments, I will suggest measurements of the cAMP value should be considered for functional readout of the mutants.

All the pharmacological have been repeated using stable cell lines expressing each mutant or the wild type receptor to give better reproducibility compared to the previously presented data obtained after transient transfection. It has also been normalised to cell surface expression. The data are in Figure 3.

5. In line 164-172, it was said that “Previous structure-activity relationship (SAR) data for mTAAR7f suggests that ligand binding is highly dependent on the ligand shape and the length of the hydrophobic chain, with aliphatic chains containing less than six carbon atoms being unable to activate the receptor”. I think the author should provide this SAR data or use a diagram to provide readers with a more intuitive view of the limitation of a certain residue on the ligand volume bound to mTAAR7f. The series of conclusions brought about by the Y132C mutation given later should be supported by corresponding data. I believe that a biochemical mutation data is necessary.

The comment about the SAR data is to provide a link between our structure and previously published data and the SAR of mTAAR7f is not the main point of this manuscript. We therefore feel that there is no need to expand on this point further, either by providing additional figures (which would be just speculation) or embarking on a mutational analysis. At the end of the day, structures of mTAAR7f would be needed to fully understand the SAR, as has been demonstrated for mTAAR9. That is beyond the scope of the current manuscript.

6. There are same issues regarding to the G protein interface. The author compared the G protein interface of mTAAR7f with other receptors and provided a barcode (fig 3b), but the corresponding mutation data is missing. It is necessary to provide mutation activity data for the corresponding site.

The most important aspect of our structure is the position of the ligand within the orthosteric binding pocket and how ligand binding induces receptor activation. A description of the G protein-coupling interface is included because many researchers are interested in it and would appreciate an overview of it in the manuscript. We are not making any claims about the relative importance of specific residues in G protein-coupling or their relative importance in, for example, biased signalling to other G proteins. Therefore, it is unnecessary to mutate every single residue and measure their effect upon signalling. Such data will have no impact on our work describing how DMCH binds and activates TAAR7f.

7. According to the map and model provided by the author, it is not difficult to find that many side chains of residues are not placed correctly, such as R107, V209, W213, etc. Additionally, there are many electron densities that are not well explained. Therefore, it is necessary for the author to redefine the model and correct Ramachandran plots and geometric issues.

Some residues, especially in flexible loops, e.g., the ones pointed out by the reviewer, were modelled based on relatively weak density indeed. Importantly, they were better resolved in the receptor-focused map that guided us through the modelling. On top of that, geometric restraints often result in non-ambiguous conformations. Considering the exact residues pointed by the reviewer, we removed the side chains of W213 and R107 and refined the model for the flexible loop 2 containing V209. This has not significantly altered the structure and has not changed any of our conclusions.

We don't understand the comment about geometric issues and Ramachandran outliers. As seen from Extended Data Table1 (both the previous version and the current one) the statistics are good.

For the revised model we used Refmac5 as it allows to give higher priority to the density fit. The model hardly changed, mostly in the remodelled flexible regions. New model statistics can be found in Extended Data Table1.

8. To ensure the accuracy of the model, it is recommended that the author add a figure showing the density of the amino acids involved in the binding pocket and the amino acids of the G protein interface in the manuscript.

Figure 1c of the submitted manuscript contained such a figure. It is Fig 1b in the revised manuscript.

9. Unfortunately, I did not see a table summarizing structural and refinement statistics of the new structure in the manuscript, this is the most basic raw data for the Cryo-EM structure, which also worries me whether some authors of this manuscript are not from a professional structure lab (Although we know Dr. Tate is very good structural biologist).

Extended Data Table 1 in the originally submitted manuscript showed all the required data in the standard *Nature* format. It is also present in the re-submitted manuscript.

Your comment "...whether some authors of this manuscript are not from a professional structure lab' is a slanderous attack on the reputation of members of my lab. I hope you have the decency to make an apology and that it is published alongside these comments. Refereeing is a privilege and an essential mainstay in the scientific endeavour. If anger and bigotry become prevalent, then science will end up being the loser.

Minor concerns

1. The reference to Fig2d in line 129 is incorrect and does not match the text. Please correct it.

Done

2. The vitality chart displayed in Extended Data Figure 9 is not satisfactory. The error bar in the figure is too large. I think authors should repeat these experiments to make more accurate measurements.

All the pharmacological have been repeated using stable cell lines expressing each mutant or the wild type receptor to give better reproducibility compared to the previously presented data obtained after transient transfection, and normalised to cell surface expression. The data are in Figure 3.

3. The sequence comparisons in Fig1a are all based on human-derived GPCRs, whereas the main subject of this paper is the murine TAAR7f. I believe that the author need to supply this figure with members of murine TAARs, including mTAAR7f.

A new figure (Extended data Figure 1) contains two panels, one comparing human TAARs with human aminergic receptors and the other comparing human TAARs with mouse TAARs.

4. For molecular dynamics simulations such as Fig1D, Fig2E, etc., the RMSD over time needs to be represented in the form of a line graph to obtain more detailed information.

We have plotted the RMSD over time for the GPCR backbone atoms and ligand atoms in the TAAR7f simulations and those with/without mini Gs (Extended Data Fig. 9).

5. Please note that many scale bar are missing for supplementary figures.

Scale bars have been added to Extended Data Figure 2d and 2e.

Reviewer #2 (Remarks to the Author):

1. Figure 1a: I don't think this figure clearly demonstrates the point that the text puts forward. It is difficult to see the distance between the TAARs and the adrenergic receptors. Most of the tree focusses on irrelevant receptors (which are also boldly colored). Perhaps it would help to make an unrooted tree and color the receptors/branches to better highlight what the readers should pay attention to.

We have tried a number of different figures to replace the phylogenetic analysis in Fig. 1a, but they were all unsatisfactory and did not show what we wanted to convey to the reader. We have therefore deleted Fig 1a and replaced this with a detailed comparison between all the relevant sequence in Extended data Fig 1. In the text we now cite exact numbers for sequence identity to convey the difference between TAARs and ORs.

2. Extended Data Figure 9: These are primary data that are important for the main point of the manuscript and thus should be included in the main figures. Simply reporting pEC50 and Emax is not sufficient. Also, the authors should explain the reason for the differences in baseline offset for the BRET measurement across the different mutations.

Given the difficulty we found in producing reproducible data using transient transfection of cells with plasmids encoding the various receptor mutants, we made stable cell lines of them all and repeated all the measurements, which also now include determination of cell surface expression levels and the correction of the BRET data to take this into account. These data are now shown in a new main figure (Fig. 3). An expanded discussion of the data is also included (lines 143-159).

3. The title references the fact that the TAAR7f ligand used is "aversive". However, the text says that mice show "either attractive/neutral or aversive behaviour when exposed to TAAR7f ligands, such as amines found in urine (ref 13)"

It is a complicated picture for the biological activity of the various ligands to the TAAR family. To avoid any potential confusion, we have removed 'aversive' from the title.

Detailed comments

Line 63: "...the TAAR studied here,..." Please clarify whether all the TAARs are more similar to adrenergic receptors as compared with ORs, or is it just TAAR7f that is more similar to adrenergic receptors.

The sentence has been clarified (lines 63-66):

'TAARs are most similar to aminergic receptors, such as the β_2 -adrenoceptor (β_2 AR), serotonin 5-HT₄ receptor, dopamine D₁ receptor and histamine H₂ receptor, with amino acid identity between these four receptors and hTAAR9 varying between 27-32% (Extended data Fig. 1a)'

Line 66: The authors should cite original articles for TAAR behaviors (PMID: 23177478; PMID: 23624375; PMID: 30158871)

Done; references 10-13, line 69.

Line 68: “TAAR7f is a well-studied murine homolog...” This line could be interpreted as TAAR7f being the closest mouse homolog of human TAAR9, which it is not. Also it is not clear to which receptor the phrase “has well characterized agonists” refers.

The sentence has been clarified (lines 71-73):

‘The TAAR receptor studied here, murine TAAR7f, has well-characterised agonists^{11,12} and its closest human homologue is TAAR9’

Line 70: “elicit” should say “exhibit”

Changed.

Line 103: “Due to the lack of...” The authors should clarify what they mean by “the ligand was placed according to its known functional properties”. On the surface, this sounds like circular reasoning (the authors placed the ligand where they thought it should go).

The sentence has been clarified (lines 106-109):

‘The density of the ligand was clearly distinguishable (Fig. 1b) and the planar configuration of the ligand was observed by the density’s flattened oval shape. The ligand was placed within this density with the positively charged tertiary amine group adjacent to the carboxylate of Asp127^{3,32}.’

Line 281: The authors should avoid the term “worse” as it is imprecise and potentially inflammatory.

Agreed. This whole section has been revised in the light of comments from Referee 1 and also the publication of the hTAAR1 structures.

Line 281: “...but the position is in line...” The authors should clarify which position—the one they observe or the one previously published.

The sentence has been clarified (lines 304-305):

‘The positions of DMCHA in both structures are within the range of poses we observe during MD simulations.’

Reviewer #3 (Remarks to the Author):

I recommend publication of the manuscript in Nature Communications without modification.

We thank the referee for their appreciation of the merits of our manuscript

REVIEWER COMMENTS

Reviewer #1 (Remarks to the Author):

In the revised manuscript, the authors have provided additional data to address my previous question. Despite the manuscript has been greatly improved following suggestions by me and the other advisor, several issues regarding to data quality and accuracy are required to be addressed before formal publication.

1. I have carefully reviewed the validation report of 8PM2 in the RCSB PDB (www.rcsb.org), where the depositor assessed the unmasked resolution as 3.60 Å, which significantly differs from the tight mask resolution of 2.92 Å. Upon further examination, I noticed that the mask map failed to encompass the TM1 portion in the mask visualization of section 6.6.1Y of the report. This oversight led to an overestimation of the overall resolution, as the density of the TM1 portion of the receptor is poorly defined.
2. In the method of Cryo-EM data processing, the box size of particle extraction is 300 Å, but its actual box size is 372 Å in the provided map. Different box sizes will bring different results. The authors should carefully revise their data and conclusions accordingly.
3. In the method section, the authors appropriately included LMNG/CHS in the buffer during the washing and elution of the mTAAR7 receptor protein. However, in subsequent steps such as SEC, G protein and Nb35 incubation as well as the final SEC of the complex, there was no mention of LMNG/CHS addition to the buffer. This omission is puzzling considering the well-established role of detergents and CHS in stabilizing the seven-transmembrane conformations of GPCR. Could the author's oversight in adding LMNG/CHS during the latter stages of purification compromise the integrity of the receptor structure?
4. Moreover, the extensive 36-hour duration of the entire purification process raises concerns regarding the maintenance of receptor stability. Could the extended duration potentially lead to structural instability?
5. There are signs of inconsistent modelling of residues throughout the entire complex structure. The reported resolution for the overall structures 2.9 Å which is OK, however the quality of the map does not match the author's claims, particularly concerning the full map. The density of many regions is very weak, and some amino acid side chains have no density at all, but the author still built the model. I have compiled a table below listing all residues with poorly modeled side chains, suggesting that this model requires further refinement or the side chains should be deleted. And any conclusion drawn from corresponding residues should be deleted because the model is not accurate.

Receptor Ga Gβ Gy Nb35
L50 E27 E3-Q6 S8-L15 Q5
Y51 D33 R8 E17 L11
F56 R38 Q9 Q18 K43
V59 V57 A11 K20 D73
L60 K58 E12 M21 K76
C63 Q59 D20 D26 E89
L67 F208 K23 I28 R105
T70 E209 K78 K32
I72 R265 E138 E42
H74 T284 R197 S57
R76 E299 E215 E58
C91 E309 M217
F94 D310 D267

M99 E314 N268
R107 E344 D323
E110 Q390
Y119 Q392
F126 L394
F180
V209
W213
M249
K273
Y317
M321
F328

Below are side-chain display diagrams that haven't properly fit into the EM density:

6. In Fig 3, many curves such as D127A, C131A, V128A, N217A, and Y289A have not reached the plateau, so the calculated EC50 is inaccurate in Fig. 3 .

7. Compared to the data from the previous submission and data from bioRxiv version, there are some differences in the cellular activity data this time. For instance, the activity of D127A, Y132A, and W286A. Is it because the data provided last time was inaccurate, or is the data not reproducible? Additionally, the Emax of F290A was reported as -1 ± 4 , Y132A as -1 ± 7 , indicating a large error bar, which may warrant further investigation.

Minor

1. Line 609, the dot in "0.01" was mistakenly written as a comma.

Reviewer #2 (Remarks to the Author):

The response of the authors has satisfied my prior concerns. The manuscript presents a solid study on the structure of an important receptor class that will garner interest as a companion to recent papers published on the same class of receptors.

I have a few minor suggestions for the authors which will improve the text.

Line 67: "TAARs bind trace amines..."

From a sensory perspective, it is more accurate to say that the olfactory TAARs bind to volatile amines. TAAR1 binds to trace amines (some of which are not that volatile for example tyramine). The definition of trace amines is based on having low concentrations in the brain and tissues.

Line 71: "The TAAR receptor studied here..." I think many readers will be thrown off by the statement that the closest orthologue to mouse Taar7f is human Taar9 (a different orthologous gene). The reason for this is that humans don't have the Taar7 family. In other words, the authors have chosen to study a gene that humans don't have (with good reason since Taar7f has known ligands). It is not clear why pointing this out is important. I would suggest removing or clarifying this statement.

Line 293: "During the preparation of this manuscript..." The authors point out explicitly that other papers came out while the paper was in preparation and review. The tone is overtly defensive. I

understand that the authors are sensitive to the issue of primacy of publication (prompted in large part by the review process). However, I don't think this is necessary. Their study is clearly independent and valuable. I would suggest they simply say "During the preparation of the manuscript..." or simply "Recent studies have shown...".

REVIEWER COMMENTS

Reviewer #1 (Remarks to the Author):

In the revised manuscript, the authors have provided additional data to address my previous question. Despite the manuscript has been greatly improved following suggestions by me and the other advisor, several issues regarding to data quality and accuracy are required to be addressed before formal publication.

1. I have carefully reviewed the validation report of 8PM2 in the RCSB PDB (www.rcsb.org), where the depositor assessed the unmasked resolution as 3.60 Å, which significantly differs from the tight mask resolution of 2.92 Å. Upon further examination, I noticed that the mask map failed to encompass the TM1 portion in the mask visualization of section 6.6.1Y of the report. This oversight led to an overestimation of the overall resolution, as the density of the TM1 portion of the receptor is poorly defined.

We thank the reviewer for checking out the validation report. We had not noticed that the auto-generated mask did not include the N-terminal end of TM1 (about 5 residues out of 919 in the model). Following the reviewer's suggestion, we generated a mask by low-pass filtering the model by 8Å and using a soft edge of 10 px and then used this to generate the FSC curves. This mask certainly includes all of TM1 (see figure below). We uploaded the corrected FSC mask into EMDDB.

For the record, the resolution predicted from the FSC curves after using the auto-generated mask was used and the correct mask we used is 2.93 Å and 2.94 Å, which is consistent with the fact that the auto-generated mask missed 5 amino acid residues out of a total of 919 residues in the model.

A. Auto-generated mask lacking part of TM1

B. Manually-generated mask including the modelled part of TM1

C.

2. In the method of cryo-EM data processing, the box size of particle extraction is 300 Å, but its actual box size is 372 Å in the provided map. Different box sizes will bring different results. The authors should carefully revise their data and conclusions accordingly.

372 is the dimension of the box in pixels, not Å. Using box dimensions in pixels is the standard for the .mrc file format.

As seen in the consensus map's header, the box size is 372 pixels, and the pixel spacing is 0.824 Å/px. This would give roughly 300Å. However, we should indeed be more precise, so we substituted 300 Å by 307 Å (300 px*0.824 Å/px) in the methods section (line 656).

```
(base) agusach@hal:~/servalcat/maps$ header map_consensus_J3.mrc
RO image file on unit 1 : map_consensus_J3.mrc      Size=      201090 K

Number of columns, rows, sections .....      372      372      372
Map mode .....      2      (32-bit real)
Start cols, rows, sects, grid x,y,z ...      0      0      372      372      372
Pixel spacing (Angstroms).....      0.8240      0.8240      0.8240
Cell angles .....      90.000      90.000      90.000
Fast, medium, slow axes .....      X      Y      Z
Origin on x,y,z .....      0.000      0.000      0.000
Minimum density .....      0.0000
Maximum density .....      0.0000
Mean density .....      0.0000
tilt angles (original,current) .....      0.0      0.0      0.0      0.0      0.0      0.0
Space group,# extra bytes,itype,lens .      0      0      0      0      0
```

1 Titles :
Relion 21-Mar-23 17:01:55

3. In the method section, the authors appropriately included LMNG/CHS in the buffer during the washing and elution of the mTAAR7 receptor protein. However, in subsequent steps such as SEC, G protein and Nb35 incubation as well as the final SEC of the complex, there was no mention of LMNG/CHS addition to the buffer. This omission is puzzling considering the well-established role of detergents and CHS in stabilizing the seven-transmembrane conformations of GPCR. Could the author's oversight in adding LMNG/CHS during the latter stages of purification compromise the integrity of the receptor structure?

The omission was an error. The methods (lines 628-630) have been altered to read "...unbound G protein and Nb35 were separated from the complex by SEC on a Superdex 200 10/300 GL column (GE Healthcare) pre-equilibrated with the buffer containing 20 mM HEPES/KOH pH7.5, 150 mM KCl, 10 mM NaCl, 10 mM MgCl₂, 6.7 mM DMCHA, and 0.01/0.001% LMNG/CHS."

4. Moreover, the extensive 36-hour duration of the entire purification process raises concerns regarding the maintenance of receptor stability. Could the extended duration potentially lead to structural instability?

We have no cause for concern about the structural instability of the complex. The complex was observed to be present on the cryo-EM grids and a structure could be determined. If there was instability of the complex that caused it to fall apart, we would not see the complex by cryo-EM.

5. There are signs of inconsistent modelling of residues throughout the entire complex structure. The reported resolution for the overall structures 2.9 Å which is OK, however the quality of the map does not match the author's claims, particularly concerning the full map. The density of many regions is very weak, and some amino acid side chains have no density at all, but the author still built the model. I have compiled a table below listing all residues with poorly modeled side chains, suggesting that this model requires further refinement or the side

chains should be deleted. And any conclusion drawn from corresponding residues should be deleted because the model is not accurate.

Receptor	G α	G β	G γ	Nb35
L50	E27	E3-Q6	S8-L15	Q5
Y51	D33	R8	E17	L11
F56	R38	Q9	Q18	K43
V59	V57	A11	K20	D73
L60	K58	E12	M21	K76
C63	Q59	D20	D26	E89
L67	F208	K23	I28	R105
T7	E209	K78		K32
I72	R265	E138		E42
H7	T284	R197		S57
R76	E299	E215		E58
C91	E309			M217
F94	D310			D267
M99	E314			N268
R107	E344			D323
E110				Q390
Y119				Q392
F126				L394
F180				
V209				
W213				
M249				
K273				
Y317				
M321				
F328				

Below are side-chain display diagrams that haven't properly fit into the EM density:

It is highly disingenuous for the referee to state "...however the quality of the map does not match the author's claims..." and then justify their statement by picking out the most flexible regions of the protein structure and show the density of the amino acid side chains in some cases is poor. Of course this is the case, because some areas of proteins are flexible and multiple positions of side chains in flexible regions will result in averaging out of the signal. However, this is no justification for criticising an objectively determined value for the resolution as this is dictated by *all* the amino acid residues and not just the ones the referee wishes to choose. In addition, we would like to point out that all the residues listed by the referee are on the periphery of the protein complex. None of them are predicted to make contact to the ligand, and neither are they in regions critical for the conformation changes of the receptor or in maintaining its architecture (purple spheres on figure below).

However, despite this, we have done extensive work to incorporate all the suggestions made by the referee, and we have been through each amino acid residue in the list supplied by the referee and remodelled the structure to be more conservative, and attached a table (below) with every amino acid residue from the reviewer's list either remodelled or explained. We have included pictures of the relevant density for the residues in the receptor (consensus map and receptor-focused map) and G protein.

Please note that the modelling we did on the receptor was performed on the density derived from the focussed classification (as described in the Methods section). The referee appeared to have used the density for the whole complex and in some instances there are differences, as is obvious by the fact that the focus refinement resulted in better resolution for the receptor portion by 0.3 Å (lines 103-106) and visibly better density.

Residue	Consensus map	Receptor-focused map	Comments
Chain R L50			Remodelled as Ala
Chain R Y51			Remodelled as Ala
Chain R F56			Remodelled as Ala

Chain R V59			Can be modelled based on receptor-focused map, the most abundant rotamer used
Chain R L60			Resolved well on receptor-focused map, density also present on consensus map-> modelled
Chain R C63			Can be modelled based on receptor-focused map, the most abundant rotamer used
Chain R L67			Can be modelled based on receptor-focused map
Chain R T70			Can be modelled based on receptor-focused map

Chain R I72			Can be modelled based on receptor-focused map
Chain R H74			Can be modelled based on receptor-focused map
Chain R R76			Remodelled as Ala
Chain R C91			The density is indeed on the weak side. Though, we decided it is important to model the side chain to highlight the fact that the C-C bridge is not being formed. The most abundant rotamer was used

Chain R F94			Can be modelled based on receptor-focused map
Chain R M99			Remodelled as Ala
Chain R R107			Remodelled as Ala
Chain R E110			Remodelled as Ala
Chain R Y119			Remodelled as Ala
Chain R Y126			Can be modelled based on receptor-focused map Y126 and Y130 make sense

Chain R F180			Remodelled as Ala
Chain R V209			The density is in the region of a flexible loop. We did our best to model the loop to avoid reader's confusion
Chain R W213			We preferred to keep the bulky side chain here as some density indicates its position
Chain R M249			The side chain can be modelled as the density is present on both maps
Chain R K273			Remodelled as Ala

Chain R Y317			Can be modelled based on receptor-focused map
Chain R M321			Can be modelled based on receptor-focused map
Chain R F328			Remodelled as Ala
Chain A R20			Remodelled as Ala
Chain A E27		-	Remodelled as Ala
Chain A D33		-	Remodelled as Ala

Chain A R38		-	Remodelled as Ala
Chain A V57		-	Still can be modelled (see the picture)
Chain A K58		-	Remodelled as Ala
Chain A Q59		-	Remodelled as Ala
Chain A F208		-	Remodelled as Ala

Chain A E209		-	Remodelled as Ala
Chain A R265		-	We find the density good enough for modelling the side chain (see the picture)
Chain A T284		-	We find the density good enough for modelling the side chain (see the picture)
Chain A E299		-	Remodelled as Ala
Chain A E309		-	The density is not the strongest, but it is present. We decided to model it.

Chain A D310		-	Same as E309, we decided to model here
Chain A E314		-	Remodelled as Ala
Chain A E344		-	We decided to model this side chain as most of the density is there
Chain A Q390		-	We find the density good enough for modelling the side chain (see the picture)
Chain A Q392		-	We find the density good enough for modelling the side chain (see the picture)

Chain A L394		-	We find the density good enough for modelling the side chain (see the picture)
Chain B E3-Q6		-	We removed the residues 3-5 and started the chain with Q6 (see teh picture)
Chain B R8		-	Remodelled as Ala
Chain B Q9		-	Remodelled as Ala
Chain B A11		-	Ala is modelled as Ala, we don't see a problem here

Chain B E12		-	The density is on the weaker side, but still allows to model the side chain (see picture)
Chain B D20		-	Remodelled as Ala
Chain B K23		-	Remodelled as Ala
Chain B K78		-	We find the density good enough for modelling the side chain (see the picture)
Chain B E138		-	We find the density good enough for modelling the side chain

Chain B R197		-	Remodelled as Ala
Chain B E215		-	Remodelled as Ala
Chain B M217		-	We find the density good enough for modelling the side chain
Chain B D267		-	Remodelled as Ala
Chain B N268		-	The density is on the weaker side, but still allows to model the side chain

Chain B D323		-	The density is on the weaker side, but still allows to model the side chain
Chain G S8-L15		-	This region is flexible and therefore less defined. We think our modelling is good enough considering this part is the furthest from the receptor and not involved in any interactions
Chain G E17		-	The density is on the weaker side, but still allows to model the side chain
Chain G K20		-	Remodelled as Ala

Chain G M21	 A molecular structure diagram showing a yellow stick model of a protein residue (M21 ALA/G) within a blue wireframe mesh representing the electron density map. The residue is oriented vertically, with its side chain pointing upwards.		Remodelled as Ala
Chain G D26	 A molecular structure diagram showing a yellow stick model of a protein residue (D26 ALA/G) within a blue wireframe mesh. The residue is oriented vertically, with its side chain pointing upwards.		Remodelled as Ala
Chain G I28	 A molecular structure diagram showing a yellow stick model of a protein residue (I28) within a blue wireframe mesh. The residue is oriented vertically, with its side chain pointing upwards.		Fits the density
Chain G K32	 A molecular structure diagram showing a yellow stick model of a protein residue (K32 LYS/G) within a blue wireframe mesh. The residue is oriented vertically, with its long side chain extending upwards.		We decided to keep the side chain on the far end of the gamma subunit, not forming any interactions
Chain G E42	 A molecular structure diagram showing a yellow stick model of a protein residue (E42) within a blue wireframe mesh. The residue is oriented vertically, with its side chain pointing upwards.		We decided to keep the side chain on the far end of the gamma subunit, not forming any interactions

Chain G S57			We decided to keep the side chain on the far end of the gamma subunit, not forming any interactions
Chain G E58			We decided to keep the side chain on the far end of the gamma subunit, not forming any interactions
Chain N Q5			The synthetic stabilizing nanobody is a subject of the study, we see no harm in placing side chains into weaker density there
Chain N L11			Fits the density
Chain N K43			The density is on the weaker side, but still allows to model the side chain

Chain N D73			The density is on the weaker side, but still allows to model the side chain
Chain N K76			The density is on the weaker side, but still allows to model the side chain
Chain N E89			The density is on the weaker side, but still allows to model the side chain
Chain N R105			The density is on the weaker side, but still allows to model the side chain

There are no major changes in the new structure. There are some slight changes around the orthosteric binding pocket where Tyr316 which was predicted to make a weak contact with the ligand is now 4.0 Å from the ligand and therefore beyond the threshold we use for saying it is making a van der Waals interaction (3.9 Å). In contrast, Cys131 was 4.0 Å from the ligand but is now 3.9 Å from the ligand and is therefore now included in the side chains that make contact. These changes are reflected in Figure 2c. As we had already discussed these sidechains in the context of the MD simulations and the mutagenesis data, none of our conclusions have changed.

6. In Fig 3, many curves such as D127A, C131A,V128A, N217A, and Y289A have not reached the plateau, so the calculated EC₅₀ is inaccurate in Fig. 3 .

The ligand is hydrophobic and has limited solubility, so it is not possible to reach higher concentrations that would give a plateau. We agree that this means that the calculated EC₅₀ is less accurate than would be the case if a plateau had been reached. We have stated this in the Methods section (lines 806-808):

“The maximum possible final concentration of DMCHA that was reproducibly attainable was 1 mM due to its relative insolubility in aqueous buffers.”

The error bars accurately reflect the uncertainties in the values for any reader to see. These errors do not impact any of our interpretations of the data or our conclusions with respect to the mechanism of ligand activation of the receptor.

7. Compared to the data from the previous submission and data from bioRxiv version, there are some differences in the cellular activity data this time. For instance, the activity of D127A, Y132A, and W286A. Is it because the data provided last time was inaccurate, or is the data not reproducible? Additionally, the E_{max} of F290A was reported as -1 ± 4 , Y132A as -1 ± 7 , indicating a large error bar, which may warrant further investigation.

As described in our previous reply to referees, the difference in the current set of data is that the level of expression of the receptors in the stable cell lines is higher and more reproducible than by transient transfection. We therefore regard the current results as a more accurate reflection of the activity of the mutants.

The absolute values of the error bars for the E_{max} values for F290A and Y132A are not significantly different from any of the other mutants that range between 2 and 31, with the values for the WT, D127A, Y289A and V312A all being larger. We do not understand why the referee thinks that there is anything untoward about the error bars for F290A and Y132A or why they may warrant further investigation. We of course take into account the magnitude of all the errors when interpreting our data and drawing conclusions on the mechanism of ligand activation of the receptor.

Minor

1. Line 609, the dot in "0.01" was mistakenly written as a comma.

Changed.

Reviewer #2 (Remarks to the Author):

The response of the authors has satisfied my prior concerns. The manuscript presents a solid study on the structure of an important receptor class that will garner interest as a companion to recent papers published on the same class of receptors.

I have a few minor suggestions for the authors which will improve the text.

Line 67: "TAARs bind trace amines..."

From a sensory perspective, it is more accurate to say that the olfactory TAARs bind to volatile amines. TAAR1 binds to trace amines (some of which are not that volatile for example tyramine). The definition of trace amines is based on having low concentrations in the brain and tissues.

Thank you for the correction. We have altered the sentence to read:

"TAARs bind volatile amines^{3,4} that are typically small molecules formed by the decarboxylation of amino acids¹⁰"

Line 71: "The TAAR receptor studied here..." I think many readers will be thrown off by the statement that the closest orthologue to mouse Taar7f is human Taar9 (a different orthologous gene). The reason for this is that humans don't have the Taar7 family. In other words, the authors have chosen to study a gene that humans don't have (with good reason

since Taar7f has known ligands). It is not clear why pointing this out is important. I would suggest removing or clarifying this statement.

In structural biology we like to know the similarity between amino acid sequences of proteins for which we have a structure e.g. between mTAAR7f and human TAARs for which there are no structures. When the sequences show high similarity, then there is a high likelihood that the structures have the same fold and that the overall mechanism is conserved. This is why we have made the statement and used the term 'homologue' to describe the relationship between the two sequences to avoid any implication that they belong to the same sub-families of TAARs (as is suggested by the word 'orthologue'). As we feel this is useful information in understanding the overall implications of the mTAAR7f structure (and related structures) in understanding their mechanism, we would like to keep this statement in. We have modified it to improve its clarity:

“The TAAR receptor studied here, mTAAR7f, has well-characterised agonists^{14,15} and its closest human homologue based on amino acid sequence is hTAAR9 (sequences are 71% identical; Extended Data Fig. 1b), implying significant conservation of their structures.”

Line 293: “During the preparation of this manuscript...” The authors point out explicitly that other papers came out while the paper was in preparation and review. The tone is overtly defensive. I understand that the authors are sensitive to the issue of primacy of publication (prompted in large part by the review process). However, I don't think this is necessary. Their study is clearly independent and valuable. I would suggest they simply say “During the preparation of the manuscript...” or simply “Recent studies have shown...”.

As suggested, we have simplified the sentence to read:

“During the preparation of the manuscript structures were published of mTAAR9²⁵ and hTAAR1^{26,27}.”

REVIEWERS' COMMENTS

Reviewer #1 (Remarks to the Author):

There is a discrepancy in the volume data values of the local resolution map compared to that in Extended Data Figure 2f, particularly in the ECL region. It is recommended that the local resolution map need to be updated to more precisely illustrate the actual local resolution in the extended data figure.

Reviewer #2 (Remarks to the Author):

The authors have appropriately responded to my comments. This is a timely paper that should be published soon.

Reviewer #1 (Remarks to the Author):

There is a discrepancy in the volume data values of the local resolution map compared to that in Extended Data Figure 2f, particularly in the ECL region. It is recommended that the local resolution map need to be updated to more precisely illustrate the actual local resolution in the extended data figure.

The local resolution map in Sup Fig 2g has been updated as suggested.

Reviewer #2 (Remarks to the Author):

The authors have appropriately responded to my comments. This is a timely paper that should be published soon.

Thank you.